# A ligand-insensitive UNC5B splicing isoform regulates angiogenesis by promoting apoptosis

Davide Pradella [1,2], Gianluca Deflorian [3,4], Alex Pezzotta[5], Anna Di Matteo [1], Elisa Belloni [1], Daniele Campolungo[1,17], Andrea Paradisi [6], Mattia Bugatti[7], William Vermi [7,8], Matteo Campioni[9], Antonella Chiapparino [9,18], Luigi Scietti[9], Federico Forneris [9], Costanza Giampietro[3,19], Nina Volf[10], Michael Rehman[10,20], Serena Zacchigna [10,11], Maria Paola Paronetto [12,13], Anna Pistocchi[5], Anne Eichmann[14,15,16], Patrick Mehlen[6] & Claudia Ghigna [1✉]

The Netrin-1 receptor UNC5B is an axon guidance regulator that is also expressed in endothelial cells (ECs), where it finely controls developmental and tumor angiogenesis. In the absence of Netrin-1, UNC5B induces apoptosis that is blocked upon Netrin-1 binding. Here, we identify an UNC5B splicing isoform (called UNC5B-Δ8) expressed exclusively by ECs and generated through exon skipping by NOVA2, an alternative splicing factor regulating vascular development. We show that UNC5B-Δ8 is a constitutively pro-apoptotic splicing isoform insensitive to Netrin-1 and required for specific blood vessel development in an apoptosis-dependent manner. Like NOVA2, UNC5B-Δ8 is aberrantly expressed in colon cancer vasculature where its expression correlates with tumor angiogenesis and poor patient outcome. Collectively, our data identify a mechanism controlling UNC5B's necessary apoptotic function in ECs and suggest that the NOVA2/UNC5B circuit represents a post-transcriptional pathway regulating angiogenesis.

[1] Istituto di Genetica Molecolare "Luigi Luca Cavalli-Sforza", Consiglio Nazionale delle Ricerche, Pavia, Italy. [2] Department of Biology and Biotechnology, University of Pavia, Pavia, Italy. [3] IFOM, The FIRC Institute of Molecular Oncology, Milan, Italy. [4] Cogentech, Società Benefit, Milan, Italy. [5] Department of Medical Biotechnology and Translational Medicine, University of Milan—LITA, Segrate, Italy. [6] Apoptosis, Cancer and Development Laboratory - Equipe labellisée "La Ligue", LabEx DEVweCAN, Centre de Recherche en Cancérologie de Lyon, INSERM U1052-CNRS UMR5286, Université de Lyon, Université Claude Bernard Lyon1, Centre Léon Bérard, Lyon, France. [7] Department of Molecular and Translational Medicine, University of Brescia, Brescia, Italy. [8] Department of Pathology and Immunology, Washington University School of Medicine, St. Louis, MO, USA. [9] The Armenise-Harvard Laboratory of Structural Biology, Department of Biology and Biotechnology, University of Pavia, Pavia, Italy. [10] Cardiovascular Biology Laboratory, International Centre for Genetic Engineering and Biotechnology (ICGEB), AREA Science Park, Padriciano 99, Trieste, Italy. [11] Department of Medical, Surgical and Health Sciences, University of Trieste, Trieste, Italy. [12] Laboratory of Molecular and Cellular Neurobiology, IRCCS Santa Lucia Foundation, Rome, Italy. [13] Department of Movement, Human and Health Sciences, University of Rome "Foro Italico", Rome, Italy. [14] Cardiovascular Research Center, Yale University School of Medicine, Department of Internal Medecine Cardiology, New Haven, CT, USA. [15] INSERM U970, Paris Center for Cardiovascular Research (PARCC), Paris, France. [16] Department of Cellular and Molecular Physiology, Yale University School of Medicine, New Haven, CT, USA. [17] Present address: IRCCS San Raffaele Scientific Institute, Division of Genetics and Cell Biology, Milan, Italy. [18] Present address: Heidelberg University Biochemistry Center (BZH), INF 328, Heidelberg, Germany. [19] Present address: Swiss Federal Laboratories for Materials Science and Technology, Experimental Continuum Mechanics EMPA, Dübendorf, Switzerland. [20] Present address: Yale University School of Medicine, Section of Nephrology, New Haven, CT, USA. ✉email: Claudia.Ghigna@igm.cnr.it

**B**lood vessels and nerves share a high degree of anatomical similarity, hence the term neurovascular link[1]. Several axon guidance regulators are also expressed by endothelial cells (ECs) and control blood vessel morphogenesis[2]. These molecules, which have a dual role in both the nervous and the vascular system, have been termed angioneurins[1].

A key angioneurin is the Unc-5 Netrin Receptor B (UNC5B)[3,4]. UNC5B is a type I transmembrane protein and contains two immunoglobulin-like (Ig) domains and two thrombospondin type I (TSP) domains in its extracellular region, whereas its cytoplasmic portion consists of a ZU5 domain (domain present in Zonula Occludens 1 and UNC5 family), a DCC-binding motif-containing domain, and a well-characterized death domain (DD)[5].

In developing blood vessels, UNC5B is present in tip cell filopodia of endothelial sprouts, which closely resemble axonal growth cones. There, it mediates the repulsive responses of Netrin chemotropins[4,6]. In ECs, UNC5B acts also as a dependence receptor, namely it induces apoptosis in the absence of its ligand Netrin-1, whereas upon Netrin-1 binding it prevents apoptosis, thus finely tuning developmental angiogenesis[5,7,8].

Although the molecular mechanisms regulating UNC5B-mediated apoptosis are still poorly known, they are fundamental to understand important aspects of developmental as well as tumor angiogenesis. Indeed, a correct balance between proapoptotic and antiapoptotic signals is critical to maintain blood vessel integrity[9–11]. While several proangiogenic factors are known to suppress apoptosis[12], induction of apoptosis is a key determinant during capillary formation and remodeling[13,14], and its inhibition reduces in vivo angiogenesis[14]. Recently, EC death has been also described as an important mechanism that sustains cancer progression[15,16].

Alternative splicing (AS) is a post-transcriptional process affecting nearly all human protein-coding genes. From a single primary transcript (pre-mRNA) it generates multiple mature mRNAs, producing protein isoforms with different structures, functions, or localization[17,18]. AS events, such as inclusion or skipping of specific exons, are controlled by the concerted action of splicing regulatory factors (SRFs) binding to cis-acting motifs located either in the regulated exons or within their flanking intronic sequences[19]. AS regulates key cellular processes during organ development, where a multiple of splicing switches are required for the acquisition of cell identity in adult tissues, including erythropoiesis[20], immune system maturation[21], spermatogenesis[22], neurogenesis[23], and myoblast differentiation[24]. The relevance of AS is also underscored by the observation that its deregulation leads to several human diseases, including cancer[25,26].

The current knowledge on the role of AS during vascular development and angiogenesis is still limited. The Neuro-oncological ventral antigen 2 (NOVA2) is a tissue-restricted SRF expressed by ECs of the blood vessels where it orchestrates vascular morphogenesis[27–29] and lymphatic EC specification[30]. NOVA2 is involved in multiple diseases. In addition to being a target in several neurodegenerative disorders[31,32], NOVA2 expression levels are specifically upregulated in tumor ECs, while are not detectable in other cell types within the tumor tissue[28,33]. In particular, NOVA2 overexpression also correlates with reduced overall survival in ovarian cancer patients[28], suggesting that altered vascular NOVA2-mediated AS programs contribute to phenotypic and functional abnormalities that characterize tumor blood vessels[34].

Here, we describe an UNC5B splicing isoform generated specifically in ECs as the result of a direct NOVA2-induced skipping of exon 8, which we named UNC5B-Δ8. We found that UNC5B-Δ8 is unable to transduce the Netrin-1 prosurvival signal and regulates blood vessels formation in an apoptosis-dependent manner. Moreover, UNC5B-Δ8 is aberrantly expressed by ECs of human colon cancer vasculature, correlating with tumor angiogenesis and poor patient survival. Collectively, our results strongly support a role for NOVA2 in controlling the dependence receptor function of UNC5B in ECs and suggest that the NOVA2/UNC5B axis may represents a post-transcriptional pathway relevant to both developmental and tumor angiogenesis.

## Results

**Identification of an *UNC5B* isoform generated by AS in the endothelium.** For the majority of protein isoforms generated through AS in ECs, functional role remains obscure[35]. AS is particularly frequent in cell-surface molecules and receptors, thus potentially affecting interactome networks as well as downstream signaling pathways[36].

Given its importance in EC biology[7,8], we investigated *UNC5B* AS events in diverse vertebrate cell and tissue types, or developmental stages, by using the Vertebrate Alternative Splicing and Transcription Database (VastDB)[37]. We found a single AS event conserved in human (HsaEX0069379), mouse (MmuEX0050632), and chicken (GgaEX0006169). This event affects *UNC5B* exon 8, whose exclusion from the mature mRNA generates a shorter transcript herein named *UNC5B-Δ8*. Intriguingly, among several SRFs, we found a strong significant inverse correlation ($r = -0.6849$; $P$ value $< 0.0001$) between the inclusion of *UNC5B* exon 8 (Percent Spliced-In, PSI) and the expression of *NOVA2*, an important regulator of vascular development[27] (Supplementary Fig. 1a). Accordingly, by RNA-seq we found that *Unc5b-Δ8* production is one of the main AS events regulated by NOVA2 in murine ECs (Fig. 1a-b). Thus, we decided to investigate the relevance of *UNC5B* AS regulation in the endothelium.

By RT-PCR analysis using RNA extracted from NOVA2 knockdown or overexpressing mouse ECs (moEC), we confirmed that NOVA2 depletion reduced the expression of *Unc5b-Δ8* (Fig. 1c), whereas an opposite effect was obtained by forced NOVA2 expression (Supplementary Fig. 1b). The NOVA2-dependent production of *Unc5b-Δ8* was also verified by performing NOVA2 silencing in other two distinct immortalized murine EC lines (Fig. 1d and Supplementary Fig. 1c). Similar results were obtained also using human ECs (Fig. 1e and Supplementary Fig. 1d). We have previously shown that NOVA2 expression is modulated by ECs density, being significantly higher in confluent compared to sparse ECs[27]. Here, we found a similar behavior also for *UNC5B-Δ8* expression (Supplementary Fig. 2a-c).

*Additionally, UNC5B-Δ8* and *NOVA2* levels increased during in vitro angiogenesis, as evaluated by a tube formation assay on Matrigel (Fig. 1f). Expression of *COL4A1*, a known marker of endothelial differentiation[38], also increased during the same time frame (Supplementary Fig. 1e).

We extended the positive correlation between *Unc5b-Δ8* and *Nova2* expression also to the vascular endothelium in vivo using freshly isolated ECs from mouse lungs (Fig. 1g). While *Unc5b* transcripts containing exon 8 (*Unc5b-FL*) are abundantly expressed in the total colon, we found negligent *Unc5b-Δ8* expression in the total colon but a marked expression in freshly purified ECs (Fig. 1g).

Since vessels in the heart undergo significant remodeling perinatally[39], we also interrogated cardiac ECs and found a progressive increase in *Unc5b-Δ8* and *Nova2* expression during mouse heart vasculature development (Fig. 1h).

Collectively, our results indicate that the AS profile of *UNC5B* is regulated in ECs in a NOVA2-dependent manner.

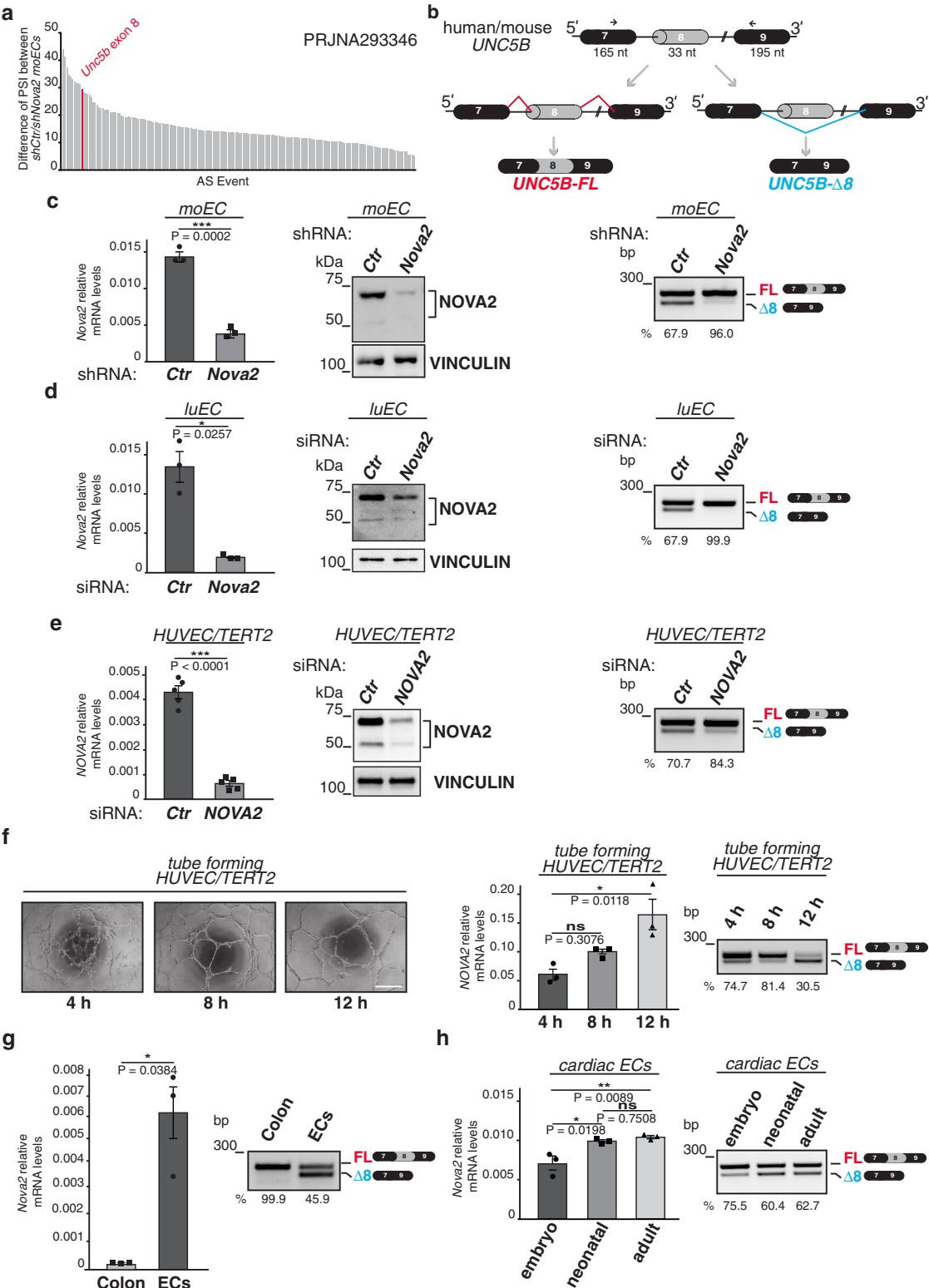

**NOVA2 regulates the skipping of *Unc5b* exon 8 directly.** NOVA2 specifically binds clusters of YCAY (Y = C/U) sequences (defined as >3 YCAY sites within 45 nucleotides) on its pre-mRNA targets[40]. Thus, we analyzed mouse *Unc5b* pre-mRNA sequence comprised between exons 7 and 9 using RBPmap (http://rbpmap.technion.ac.il)[41] (see Methods). The top 7 RNA-binding motifs (Z-score ≥ 3.50) were YCAY tetranucleotides located in the intronic region upstream of *Unc5b* exon 8 (Fig. 2a; Supplementary Fig. 3a). SpliceAid 2 (http://www.introni.it/spliceaid.html)[42] reported similar results (Supplementary Fig. 3b). Importantly, the YCAY cluster position was consistent with *Unc5b* splicing regulation, given the ability of NOVA2 to induce exon skipping when bound upstream to the regulated AS exon[40]. Notably, five out of seven of the top motifs identified in mouse *Unc5b* pre-mRNA were conserved in human *UNC5B* pre-mRNA (Fig. 2a; Supplementary Fig. 3c). To determine whether

**Fig. 1 Inclusion of *UNC5B* exon 8 is regulated in ECs. a** Differentially spliced exonic events in NOVA2-depleted moEC compared to control moEC (NCBI BioProject: PRJNA293346). Absolute difference of Percent Spliced-In (PSI) between means is shown. *Unc5b* exon 8 AS event is indicated in red. **b** Schematic representation of the mouse and human *UNC5B* genomic region with the AS exon 8 of 33 nt (gray). Black boxes = constitutive exons; lines = introns. Two different mRNAs result from the inclusion (red lines) or skipping (blue lines) of exon 8. Arrows indicate primers used for RT-PCR. **c** Left: RT-qPCR (relative to *Ubb*) and immunoblotting of NOVA2 expression levels in NOVA2-depleted moEC. The anti-NOVA2-specific antibody recognizes two immunoreactive bands at 50–55 kDa and 70–80 kDa, as previously reported[27, 31]; VINCULIN as loading control. Right: RT-PCR of *Unc5b* exon 8 AS profile in the above ECs. n = 3. **d** *Nova2* mRNA (relative to mouse *Ubb* or human *ACTB*) and protein expression levels, and AS analysis of *Unc5b* exon 8 in another murine EC line (luEC) (n = 3) and **e** in human immortalized ECs (HUVEC/TERT2) transfected with a siRNA against *Nova2* or with a control siRNA. VINCULIN as loading control. n = 5. **f** Left: in vitro angiogenesis assay of HUVEC/TERT2 plated on Matrigel-coated plates. Scale bar: 0.5 mm. Right: *NOVA2* mRNA expression levels (relative to *B2M*) and *UNC5B* exon 8 AS during differentiation of HUVEC/TERT2 and formation of capillary tube-like structures on Matrigel. h = hours after seeding. n = 3. **g** Analysis, by RT-qPCR, of *Nova2* mRNA expression levels (relative to *Ubb*) and, by RT-PCR, of *Unc5b* exon 8 splicing in freshly purified ECs from mouse lung and colon tissue. n = 3. **h** *Nova2* mRNA expression levels (relative to *Ubb*) and *Unc5b* exon 8 AS in primary ECs from mouse hearth (cardiac ECs) at different developmental stages: embryonic (E16–E18); neonatal (P0-P1); adult (P60). n = 3. The percentage of exon inclusion is shown under each gel. n = biologically independent experiments. Two-tailed Student's *t*-test or one-way ANOVA for multiple comparisons; Error bars indicate ±SEM. Exact *P* values are indicated: *$P < 0.05$; **$P < 0.01$; ***$P < 0.001$; ns not significant.

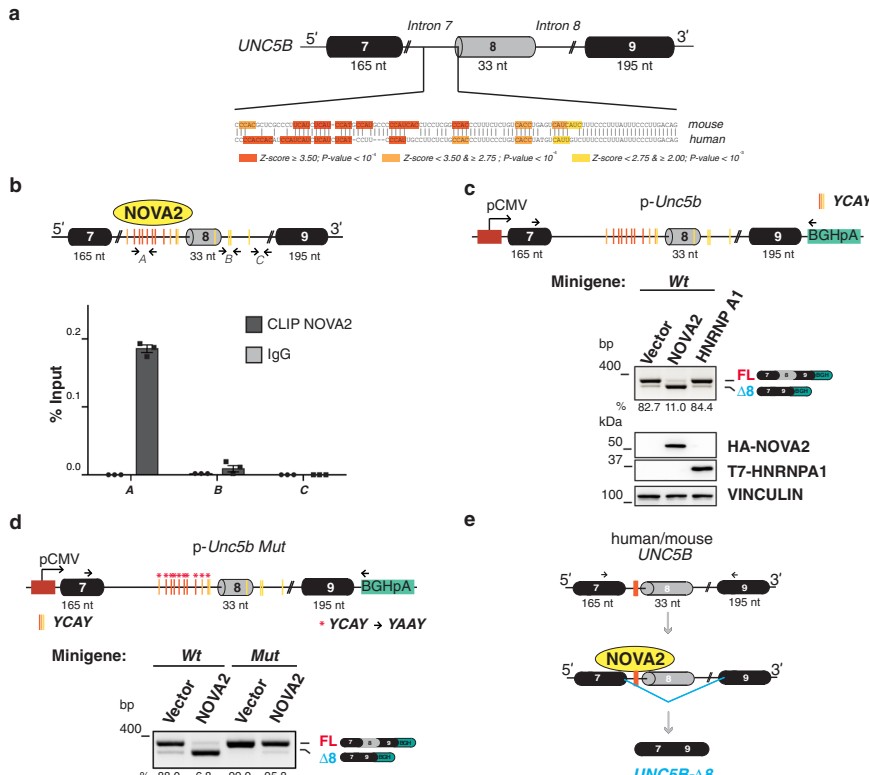

**Fig. 2 NOVA2 directly binds to *Unc5b* pre-mRNA. a** Schematic representation of mouse and human *UNC5B* pre-mRNAs. AS exon 8 in gray. Constitutive exons 7 and 9 in black. Introns 7 and 8 are also indicated (thin lines). YCAY sites, identified by using the RBPmap tool within 200 nt from the exon–intron junctions, are indicated in orange, pale orange, or yellow bars depending on their Z-score and *P* values (calculated by RBPmap tool). **b** Upper: schematic representation of mouse *Unc5b* as in **a**. Arrows indicate primers used for RT-qPCR of NOVA2 immunoprecipitated RNAs. Lower: RT-qPCR of NOVA2 CLIP experiments in moEC. Amplification of immunoprecipitated RNAs with primers annealing to the YCAY cluster in intron 7 (*A*) and, as negative controls, within intron 8 (*B* and *C*). Error bars indicate ±SD calculated from n = 3 independent experiments. **c** Upper: the mouse *Unc5b* genomic region encompassing exons 7, 8, and 9, cloned into the pcDNA3.1(+) vector to generate the p-*Unc5b* wild-type (*Wt*) minigene. Arrows indicate primers used for splicing analysis of the transcripts generated from the minigene. Boxes = exons; thin lines = introns; pCMV = promoter; BGHpA = polyadenylation sequence. Colored vertical bars as in **a**. Lower: p-*Unc5b* *Wt* co-transfected in moEC with either NOVA2-HA (NOVA2), T7-HNRNPA1 (HNRNPA1), or empty vector (Vector). RT-PCR analysis of p-*Unc5b* splicing is shown. Ectopic expression of NOVA2 and HNRNPA1 were confirmed by immunoblotting with anti-HA and anti-T7 antibodies, respectively. VINCULIN as loading control. **d** Splicing assay with p-*Unc5b* *Mut* in moEC. Red asterisks indicate the mutated NOVA2 binding sites (YCAY repeats were replaced with YAAY). **e** NOVA2 (yellow circle) promotes *UNC5B* exon 8 skipping by directly binding to the YCAY cluster (orange bar) located in the intronic region upstream exon 8. The percentage of exon inclusion is shown below each gel. At least three independent biological replicates were analyzed for each experiment.

the YCAY cluster in *Unc5b* intron 7 represented a bona fide NOVA2 binding site, we performed UV crosslinking and immunoprecipitation (CLIP), which detects direct protein-RNA interactions in living cells[43]. RNA of UV crosslinked ECs was immunoprecipitated using anti-NOVA2 or control antibodies. RNAs bound by NOVA2 were then analyzed by RT-qPCR with primers spanning either the YCAY cluster in *Unc5b* intron 7 (primers A) or intron 8 (primers B and C), used as controls

(Fig. 2b). We found strong NOVA2 enrichment on the endogenous *Unc5b* pre-mRNA at the level of the YCAY cluster in intron 7 (Fig. 2b).

To further investigate the role of NOVA2 in regulating *Unc5b* splicing, we performed a splicing assay by using an *Unc5b* minigene encompassing exons 7, 8, and 9 of the mouse *Unc5b* gene along with the complete intervening intronic sequences (Fig. 2c). The minigene was co-transfected in moEC with plasmids encoding either NOVA2 or HNRNPA1, an SRF abundantly expressed by ECs[44], for which binding sites were not detected in the sequence of mouse *Unc5b* by our RBPmap analysis (Supplementary Fig. 3a). An empty plasmid was used as a control. As showed in Fig. 2c, skipping of *Unc5b* exon 8 was increased by NOVA2 overexpression. In contrast, HNRNPA1 overexpression had no effect, supporting a NOVA2-specific effect. We also generated a mutated *Unc5b* minigene (p-*Unc5b Mut*) in which YCAY motifs upstream exon 8 were replaced with YAAY, a sequence not recognized by NOVA2[45]. As showed in Fig. 2d, mutations in the YCAY cluster reduced *Unc5b-Δ8* production when NOVA2 was overexpressed (Fig. 2d). We also performed an in vitro RNA pull-down to further assess the sequence specific association of NOVA2 to *Unc5b* YCAY motifs in intron 7. As showed in Supplementary Fig. 4a, NOVA2 was able to interact with *Unc5b* intron 7 riboprobe containing the YCAY cluster, but not with its mutated version (Supplementary Fig. 4a).

Downstream *Unc5b* exon 8, our RBPmap analysis indicated the presence of putative RBFOX2 binding sites, an SRF with a well-characterized role in ECs[46]. Nevertheless, silencing of RBFOX2 did not affect *Unc5b* splicing in moEC (Supplementary Fig. 4b) further supporting the specificity of our results.

In summary, our findings demonstrate that NOVA2 promotes skipping of *UNC5B* exon 8 by directly binding to the YCAY cluster located upstream of this exon (Fig. 2e).

**Unc5b-Δ8 is required for angiogenesis in vivo**. Zebrafish represents a powerful model to study vertebrate vascular development[47]. Moreover, the Nova2 RNA-binding domain is 94% identical in zebrafish and humans, and ~50% of Nova-regulated AS events are conserved from mouse to zebrafish[48]. We found that a cluster of YCAY motifs is also located upstream of the zebrafish *unc5b* exon 8 (Fig. 3a). Accordingly, we discovered that the expression of the *unc5b-Δ8* was co-regulated with *nova2* levels during zebrafish development (Fig. 3b). Morpholino-mediated *nova2* knockdown caused a reduction of *unc5b-Δ8* production, which was partially rescued by the co-injection of a morpholino-resistant *nova2* mRNA, thus supporting a Nova2-specific effect (Fig. 3c; Supplementary Fig. 5a). To validate these findings, we used *nova2* mutant fish generated by CRISPR/Cas genome editing[27]. Notably, *nova2* mutants displayed AS changes of *unc5b* exon 8 comparable to those observed in *nova2* morphants (Fig. 3d). Collectively, these results indicated that *unc5b* exon 8 splicing is regulated by Nova2 in vivo in zebrafish embryos.

At the protein level, zebrafish Unc5b shares 65% amino acid identity with the human receptor[49], with the highest percentage identity (71%) in the netrin-binding Ig domains[50]. Unc5b depletion in zebrafish causes vessel development defects in specific vascular beds. In *unc5b* morphants, inhibition of parachordal vessel (PAV), formed by secondary sprouts emerging from the posterior cardinal vein (PCV), is a highly penetrant phenotype[4,5,51] and it was attributed to dysregulated Unc5b prodeath activity[7]. As showed in Fig. 3e, we confirmed PAV defects in *unc5b* morphants (Fig. 3e). By using PAV formation as a readout of Unc5b function in vivo, we also found that co-

injection of a morpholino-resistant *unc5b-Δ8* mRNA in *unc5b* morphants showed higher efficiency in restoring Unc5b vascular activity compared to the full-length mRNA containing exon 8 (*unc5b-FL*) (Fig. 3e and Supplementary Fig. 5b-c). While injections of increasing doses of *unc5b-FL* were not able to restore PAV formation (Supplementary Fig. 5d-e), we were able to observe an increase of PAV sprouting in zebrafish embryos injected with the maximal dose tested (Supplementary Fig. 5f).

PAV growth is also strongly inhibited in *nova2* mutants at both 72 hours (hpf) and 6 days post-fertilization (dpf) (Fig. 3f and Supplementary Fig. 6a), thus indicating complete abrogation of PAV formation upon Nova2 depletion rather than a delay. The drastic reduction of *unc5b-Δ8* observed upon Nova2 depletion (Fig. 3c-d) prompted us to evaluate the ability of *unc5b-Δ8* mRNA injection to restore PAV formation also in a background where the endogenous *unc5b-FL* isoform is expressed. Remarkably, we found that *unc5b-Δ8* mRNA rescued PAV formation defects of *nova2* mutant (Fig. 3g), thus suggesting that Unc5b-Δ8, but not Unc5b-FL isoform, is required for the formation of the PAV during zebrafish development.

To further investigate the function of Unc5b-Δ8 during vascular development, we injected increasing doses of *unc5b-Δ8* mRNA in wild-type zebrafish embryos—where, differently from rescue experiments of *unc5b* morphants (Fig. 3e), both Unc5b-FL and Unc5b-Δ8 isoforms are present.

Forced expression of Unc5b-Δ8 caused several morphological defects in a dose-dependent manner, including loss of arterial-ISV formation abnormalities and vessels defects in the plexus region (Supplementary Fig. 6c-f).

Collectively, our results support the biological relevance of Unc5b-Δ8 in the formation of several vascular beds and the importance of a tight control of Unc5b-Δ8 production during zebrafish development. In particular, whereas Unc5b-Δ8 is fundamental for the formation of the PAV during zebrafish development, its forced expression is detrimental for other vasculature structures, including the ISVs and vessels in the plexus region.

**UNC5B-Δ8 prevents Netrin-1 prosurvival signaling**. Apoptosis plays a fundamental role in tissue remodeling during development through the removal of redundant cells. In the context of vascular biology, apoptosis is involved in several cases of vessel regression[11] as well as in important aspects of angiogenic vessel growth, including lumen formation[52,53], EC sprouting[54], and the control of EC number and capillary diameter during vessel maturation[55].

In the absence of Netrin-1, UNC5B induces apoptosis through its cytoplasmic death domain (DD). Instead, Netrin-1 ligand binding triggers UNC5B multimerization leading to the inhibition of UNC5B prodeath activity[8]. In this latter condition, the cytoplasmic region of UNC5B adopts a closed conformation[5] and interacts with an inactive, phosphorylated form of Death-Associated-Protein kinase (DAPk)[8]. Conversely, in the absence of Netrin-1, UNC5B adopts an open conformation that triggers DAPk dephosphorylation (active form) and induction of apoptosis[8] (Fig. 4a).

Given the importance of cell death regulation for EC functions, we investigated whether the NOVA2-dependent AS regulation of *UNC5B* influence UNC5B cell death activity. We expressed the two UNC5B AS variants, UNC5B-FL (with exon 8) and UNC5B-Δ8 (lacking amino acid sequence Asn[356]- Pro[367], which are substituted by a Thr[367] residue), fused to a C-terminal HA- or GFP-tag. As showed in Fig. 4b, both variants were equally able to reach the cell membrane of transduced HUVEC (Fig. 4b). Similar results were obtained using other overexpressing ECs (such as

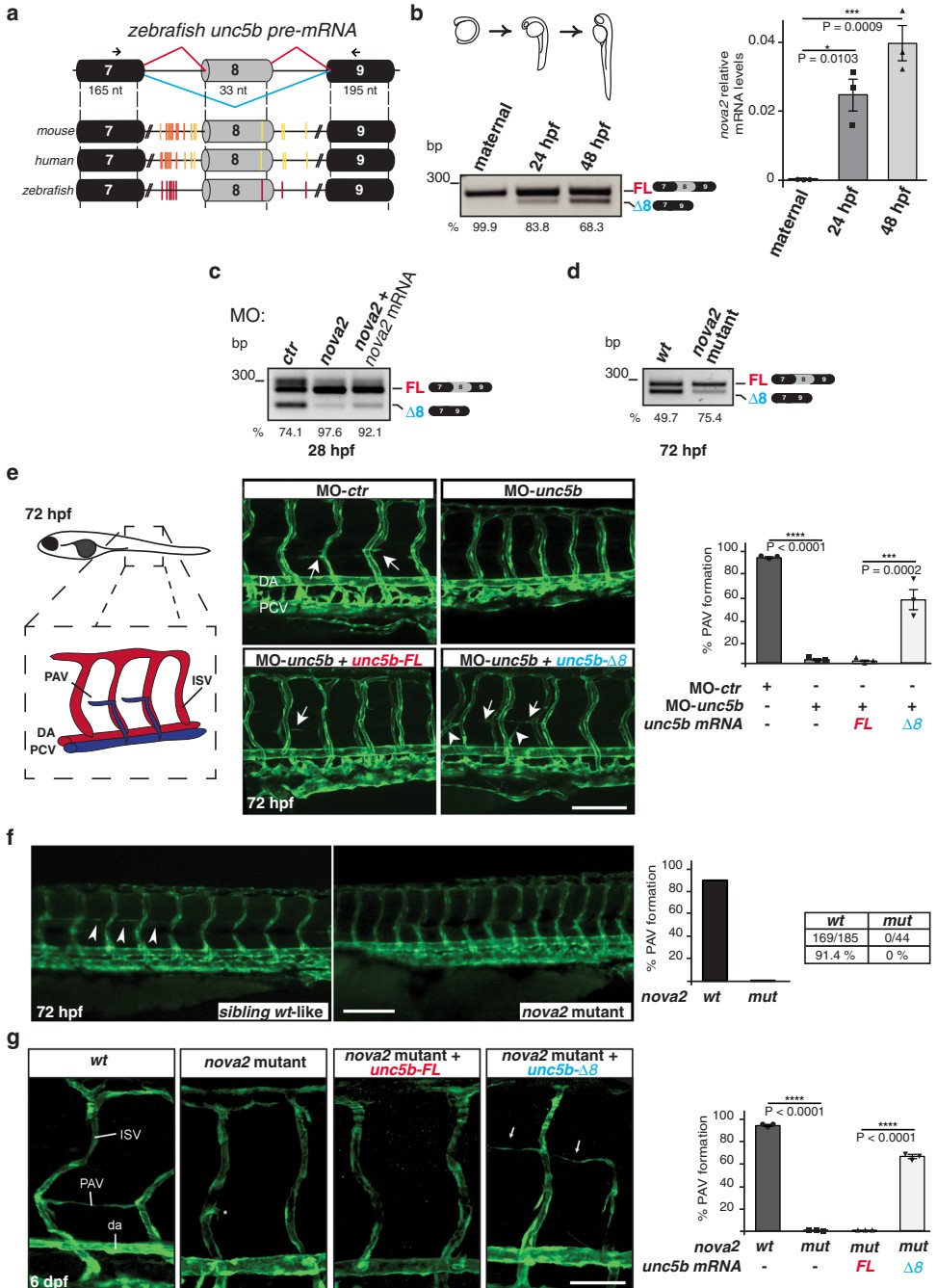

**Fig. 3 *Unc5b-Δ8* promotes PAV vessel formation in zebrafish. a** Schematic representation of mouse, human, and zebrafish *unc5b* genomic regions with the AS exon 8 in gray. Black boxes = constitutive exons; lines = introns. YCAY motifs identified with RBPmap tool are indicated as vertical bars. Arrows indicate primers for RT-PCR. **b** Splicing analysis of *unc5b* exon 8 and *nova2* mRNA expression levels in zebrafish embryos at different developmental stages: maternal, 24 h post-fertilization (hpf), and 48 hpf. **c** RT-PCR with RNA extracted from zebrafish embryos (28 hpf) injected with a control morpholino (MO-*ctr*) or a morpholino against *nova2* (MO-*nova2*). Altered AS was partially corrected by the co-injection of a morpholino-resistant *nova2* mRNA (MO-*nova2* +*nova2*). *n* = 6. **d** Compared to the wild-type (wt) organism, *nova2* mutants show a reduction of the *unc5b-Δ8* transcript (72 hpf). The percentage of exon inclusion is indicated. *n* = 3. **e** Left: scheme of zebrafish vessels; PAV parachordal vessel, DA dorsal aorta, PCV posterior cardinal vein, ISV intersegmental vessel. Central: lateral views (fluorescence) of *Tg(kdrl:GFP)* *la116* zebrafish embryos (72 hpf) injected with a control morpholino (MO-*ctr*) or with a morpholino against *unc5b* (MO-*unc5b*); *unc5b* morphants were also co-injected with morpholino-resistant zebrafish mRNAs encoding for *unc5b* AS isoforms (*unc5b-FL* or *unc5b-Δ8*). Right: quantification of embryos in which PAV is correctly formed (%). *n* = 3. Arrows indicate correctly forming PAV, whereas arrowheads indicate those unproperly forming. **f** Left: Lateral views (fluorescence) of 72 hpf wild-type (*wt*) and *nova2* CRISPR mutant zebrafish embryos. Right: Quantification of PAV formation in 72 hpf wild-type (wt) and *nova2* mutant (*mut*). N. of larvae with normal PAV out the total n. of larvae. Arrowheads indicate forming PAV. **g** Left: lateral views (fluorescence) of *wt* and *nova2* mutant embryos (6 dpf) injected with mRNAs encoding for *unc5b* AS isoforms (*unc5b-FL* or *unc5b-Δ8*). Arrows indicate forming PAV. Right: quantification of embryos in which PAV is correctly formed (%). *n* = 3. n = biologically independent experiments. One-way ANOVA for multiple comparisons; Error bars indicate ±SEM. Exact *P* values are indicated: *P < 0.05; ***P < 0.001; ****P < 0.0001. Scale bars: 50 μm in **e**, 100 μm in **f**, 25 μm in **g**.

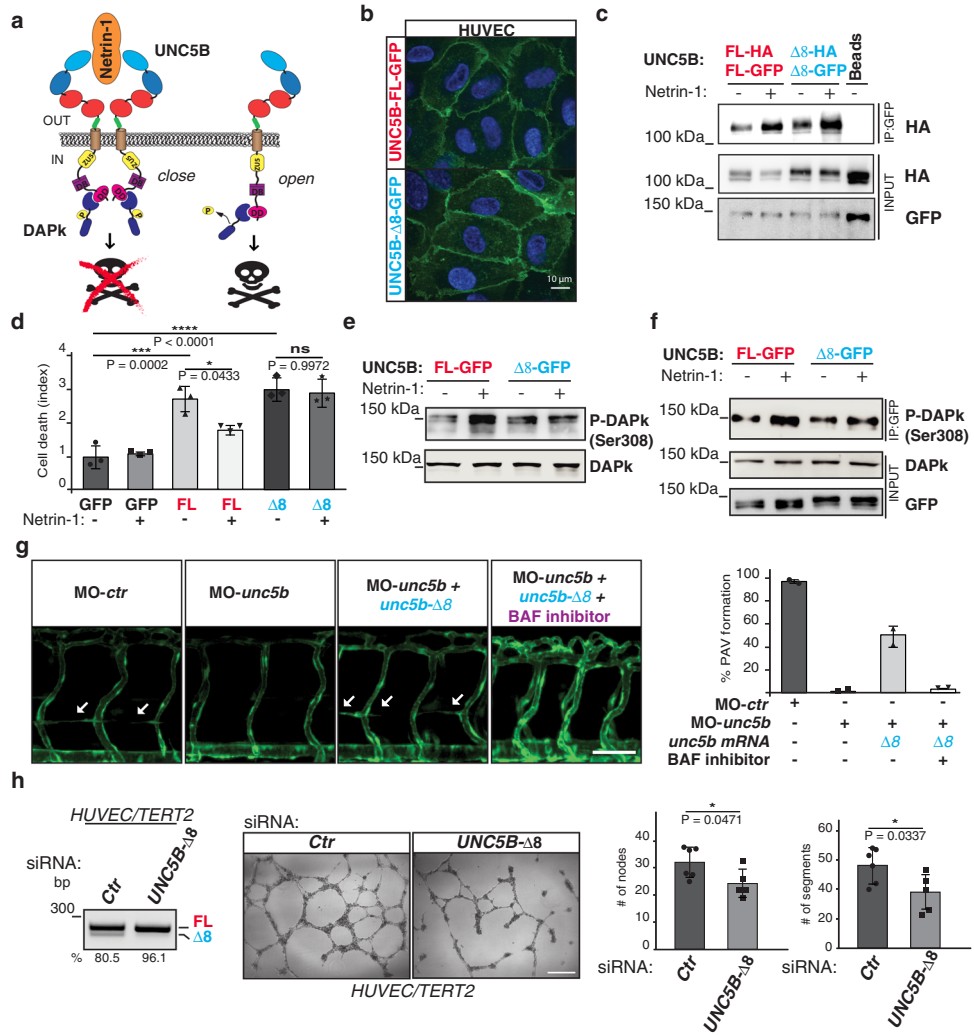

**Fig. 4 UNC5B-Δ8 prevents Netrin-1 prosurvival signaling and promotes vessel formation in an apoptotic-dependent manner. a** Schematic representation of Netrin-1/UNC5B-mediated prosurvival signal. UNC5B domains are indicated with different colors. Blue ovals: Ig-like domains; red ovals: thrombospondin-like domains (TSP1); beige cylinder: transmembrane domain (TM); yellow circle: ZU5 domain; violet cube: UPA/DCC-binding domain (DB); pink circle: death domain (DD). Region encoded by exon 8 is shown in green. Netrin-1 is in orange, whereas DAPk is in dark blue. DAPk phosphorylation at Ser308 is indicated. **b** Localization of GFP-tagged UNC5B isoforms in HUVECs. Scale bar: 10 μm. **c** Co-immunoprecipitation of UNC5B AS isoforms GFP-tagged with HA-tagged isoforms in the presence or absence of Netrin-1 (150 ng/ml). Unspecific magnetic beads (Beads) as control. **d** Vital exclusion assay in HeLa cells overexpressing UNC5B AS isoforms or GFP as control. Error bars indicate ±SD; $n = 3$. One-way ANOVA for multiple comparisons. **e** DAPk phosphorylation (Ser308) was evaluated in HeLa cells co-transfected with plasmids expressing DAPk and Unc5b isoforms with or without Netrin-1 treatment (150 ng/ml). Total DAPk as control. **f** Co-immunoprecipitation of Phospho-DAPk (Ser308) in HeLa cells transiently transfected as in **e**. **g** Left: Lateral views (fluorescence) of 72 hpf zebrafish embryos expressing the GFP under the control of the endothelial-specific promoter *kdrl* injected with a control morpholino (MO-*ctr*) or with a morpholino against *unc5b* (MO-*unc5b*); *unc5b* morphants were also co-injected with a morpholino-resistant zebrafish mRNA encoding for *unc5-Δ8* (*unc5b-Δ8*) and treated with a pan caspase inhibitor (BAF inhibitor). White arrows: forming PAV. Right: Quantification of PAV formation in the above zebrafish embryos, two different biological replicates were analyzed. $n = 2$. Error bars indicate ±SEM. Scale bar: 50 μm. **h** HUVEC/TERT2 treated with a siRNA oligo against the *UNC5B-Δ8* mRNA ($n$ 5) or a control (*Ctr*) oligo ($n = 6$) were analyzed (left) by RT-PCR for *UNC5B* exon 8 splicing and (center) for the formation of capillary tube-like structures on Matrigel. Right: quantification of nodes and segments (# per field) in the in vitro tube formation assay. Two-tailed Student's t-test. Error bars indicate ±SD. $n =$ biologically independent experiments. Exact P values are indicated: *$P < 0.05$;; ***$P < 0.001$; ****$P < 0.0001$; ns not significant.

moEC) (Supplementary Fig. 7a). Since HeLa expresses no or very little endogenous UNC5B (Supplementary Fig. 7b), we used these cells for gain-of-function studies. Both UNC5B-FL and UNC5B-Δ8 reached the cell membrane (Supplementary Fig. 7c) exposing their Netrin-binding extracellular portion (Supplementary Fig. 7d) in transiently transfected HeLa cells.

Next, using UNC5B isoforms fused to HA- or GFP-tag, we tested the ability of these receptors to dimerize in the presence of Netrin-1[56]. As showed in Fig. 4c, UNC5B-Δ8 was able to properly dimerize with other UNC5B-Δ8 receptors, as it has been reported

to occur for the UNC5B-FL isoform[56] (Fig. 4c). Using surface plasmon resonance (SPR), as previously reported[57], we measured similar Netrin-1-binding affinities for the recombinant ectodomain fragments of both UNC5B variants (Supplementary Fig. 7e). Hence, the absence of 11 amino acids (exon 8) in the extracellular portion of UNC5B does not affect the ability of UNC5B-Δ8 to form dimers or associate to its ligand Netrin-1.

Next, we tested if the absence of exon 8 could influence the ability of Netrin-1 to block UNC5B-induced cell death[7]. As showed in Fig. 4d and e, we confirmed that Netrin-1 treatment

blocks cell death induction by UNC5B-FL (Fig. 4d) and that this effect is accompanied by an increase of the phospho-DAPk (Fig. 4e), the UNC5B-mediator of apoptosis inactive in the phosphorylated state[7]. We also found that Netrin-1 treatment increased phospho-DAPk association to UNC5B-FL compared to untreated cells (Fig. 4f and Supplementary Fig. 7f). Conversely, cells expressing UNC5B-Δ8 were largely irresponsive to Netrin-1 treatment, as determined by: (i) cell death assays, (ii) levels of phospho-DAPk; and (iii) the association of the latter with the receptor (Fig. 4d-f). The confirmation of these results in ECs (such as HUVEC and moEC) (Supplementary Fig. 8a-d) strongly suggests that NOVA2-mediated splicing of UNC5B exon 8 transforms the dependence receptor UNC5B in a constitutively active proapoptotic variant.

During zebrafish development, apoptosis occurs in transient tissue- and cell-population-restricted waves[58]. To assess EC death in the developing vascular system, we performed a terminal deoxynucleotidyl transferase (TdT) dUTP Nick-End Labeling (TUNEL) assay in zebrafish embryos at different developmental stages (Supplementary Fig. 9a-b). Dying ECs were detectable in the zebrafish trunk and tail regions at 30/33 hpf, a developmental stage short before a second wave of sprouting emerges from the PCV to remodel ISVs and form the PAV that occurs at 34 hpf (Supplementary Fig. 9a-b). Notably, apoptotic cells were observed in the region of the PCV from which the PAV originates—at 30 hpf—or where the PAV would arise—at 33 hpf. Apoptotic ECs were also confirmed by cleaved Caspase-3 immunostaining (Supplementary Fig. 9c).

The ability of UNC5B to induce apoptosis is intimately connected with its role in angiogenesis[5]. Indeed, an UNC5B mutant in a constitutively open and proapoptotic conformation stimulates the formation of specific blood vessels (such as PAV) in zebrafish embryos[5]. Since we found that Unc5b-Δ8 is required for adequate PAV formation during zebrafish development (Fig. 3e) and its overexpression induced EC death in vivo (Supplementary Fig. 8e), we tested whether apoptosis is required for PAV formation.

Unc5b depletion reduced detectable EC death in zebrafish embryos, which was restored by unc5b-Δ8 mRNA injection. Coherently, in co-injected embryos treated with different apoptosis inhibitors, we were unable to detect dying ECs detected by TUNEL assay (Supplementary Fig. 8f). Notably, inhibition of apoptosis prevented the unc5b-Δ8 mRNA-mediated rescue of PAV formation defects in unc5b morphants, without impairing other blood vessels (Fig. 4g, Supplementary Fig. 8g).

Finally, we tested UNC5B-Δ8-depleted human ECs capacity to form tube-like structures in two-dimensional matrices, a process in which both pharmacological and genetic inhibition of apoptosis prevent the formation of a mature network[14]. Notably, UNC5B-Δ8 depletion by using a small interfering RNA (Fig. 4h) or by using morpholino oligos sterically blocking NOVA2 binding on YCAY cluster located in intron 7 (Supplementary Fig. 10a-d) significantly reduces the tube formation capabilities.

In summary, our data indicate that UNC5B-Δ8 prevents Netrin-1 prosurvival signaling and that its apoptotic activity is necessary for angiogenesis.

**NOVA2/UNC5B-Δ8 axis in ECs of colon cancer patient vasculature**. Compared to normal blood vessels, NOVA2 expression is upregulated in tumor ECs of colorectal and ovarian cancers, whereas it is not detectable in other cell types within the tumor[28,33]. Moreover, high NOVA2 expression correlates with shorter overall ovarian cancer patient survival[28].

By immunohistochemistry (IHC), we confirmed NOVA2 selective upregulation in ECs of colon adenocarcinoma vasculature and other cancer types—including oral cavity and hepatocellular carcinomas—compared to adjacent normal tissues (Fig. 5a; Supplementary Fig. 11a-c and Supplementary Fig. 12a-b). Next, we performed an in situ hybridization (ISH) by using a chromogenic BaseScope assay able to specifically detect UNC5B-Δ8, but not the UNC5B-FL mRNA (Supplementary Fig. 13a-b). According to EC-restricted NOVA2 expression in cancer specimens, we found that UNC5B-Δ8 was present only in the tumor endothelium (Fig. 5a and Supplementary Fig. 13c).

Using the UALCAN web-tool (http://ualcan.path.uab.edu)[59] to compare cancer and nonpathological samples from The Cancer Genome Atlas (TCGA)[60], we found that NOVA2 was upregulated in colon cancer tissues versus not pathological counterparts (Fig. 5b). NOVA2 upregulation was also confirmed in colon cancer microarray datasets from Oncomine (http://oncomine.org/resource)[61] (Supplementary Fig. 14a-b). RNA-seq TCGA dataset also showed a significant decrease of UNC5B exon 8 inclusion in colorectal cancers compared to the normal tissues (Fig. 5c). Accordingly, a positive correlation (r = 0.406; P < 0.0001) between UNC5B-Δ8 and NOVA2 expression levels was present in the TCGA-Colon Adenocarcinoma (TCGA-COAD) dataset (Fig. 5d). We validated these findings in a small cohort of colon cancer patients by RT-PCR and RT-qPCR analysis by using RNA extracted from: (i) colon cancer specimens, (ii) normal adjacent colon tissue, and (iii) liver metastasis (Supplementary Fig. 14c-e).

A similar correlation was also found in liver hepatocarcinoma (LIHC-TCGA) and head and neck tumors (HNSC-TCGA) (Supplementary Fig. 12c-d), where our IHC analyses demonstrated EC-restricted NOVA2 expression. Expression levels of both NOVA2 and UNC5B-Δ8 were also positively correlated with the expression of a colon tumor-specific angiogenetic signature[62] (r = 0.729 and r = 0.430; P < 0.0001) (Fig. 5e and f).

Clinical data annotated in the TCGA dataset (https://www.cbioportal.org)[63] and the Human Cancer Metastasis Database (HCMDB) (https://hcmdb.i-sanger.com)[64], showed NOVA2 upregulation in colon cancers with metastasis and/or recurrences (Fig. 5g-h). Indeed, high NOVA2 expression was also associated with shorter metastasis and relapse-free survival (Supplementary Fig. 14f-g). We also found that high NOVA2 and UNC5B-Δ8 expression levels correlated with short overall survival in colon cancer patients (Fig. 5i-l).

Recent evidence has shown that tumor cells are able to induce ECs cell death, thus favoring metastasis dissemination[16]. Here, we were able to detect positive apoptotic ECs within the tumor tissue, whereas detection of apoptosis in the vasculature of the nonpathological parenchyma was rare (Supplementary Fig. 15a-b). Notably, apoptotic ECs showed a strong NOVA2 IHC signal. However, due to the difficulty of detect EC-restricted apoptotic events in such a small cellular population of the tumor microenvironment, our observations do not allow us to directly evaluate the impact of EC apoptosis in the disease progression.

Collectively, our results show that UNC5B-Δ8 and NOVA2 are upregulated in colorectal tumor vasculature. Furthermore, the correlation of UNC5B-Δ8 and NOVA2 expression with a tumor angiogenic signature is in line with the possibility that the NOVA2/UNC5B-Δ8 circuit plays a role also in the tumor vascular niche. However, further studies are required to better assess the contribution of UNC5B-Δ8 and NOVA2 to tumor vascularization and metastasis.

## Discussion

Tight regulation of the balance between proapoptotic and anti-apoptotic signals is crucial to maintain blood vessel integrity and stability[9–11]. While a number of proangiogenic factors prevent EC apoptosis[12], time- and spatial-restricted induction of apoptosis is

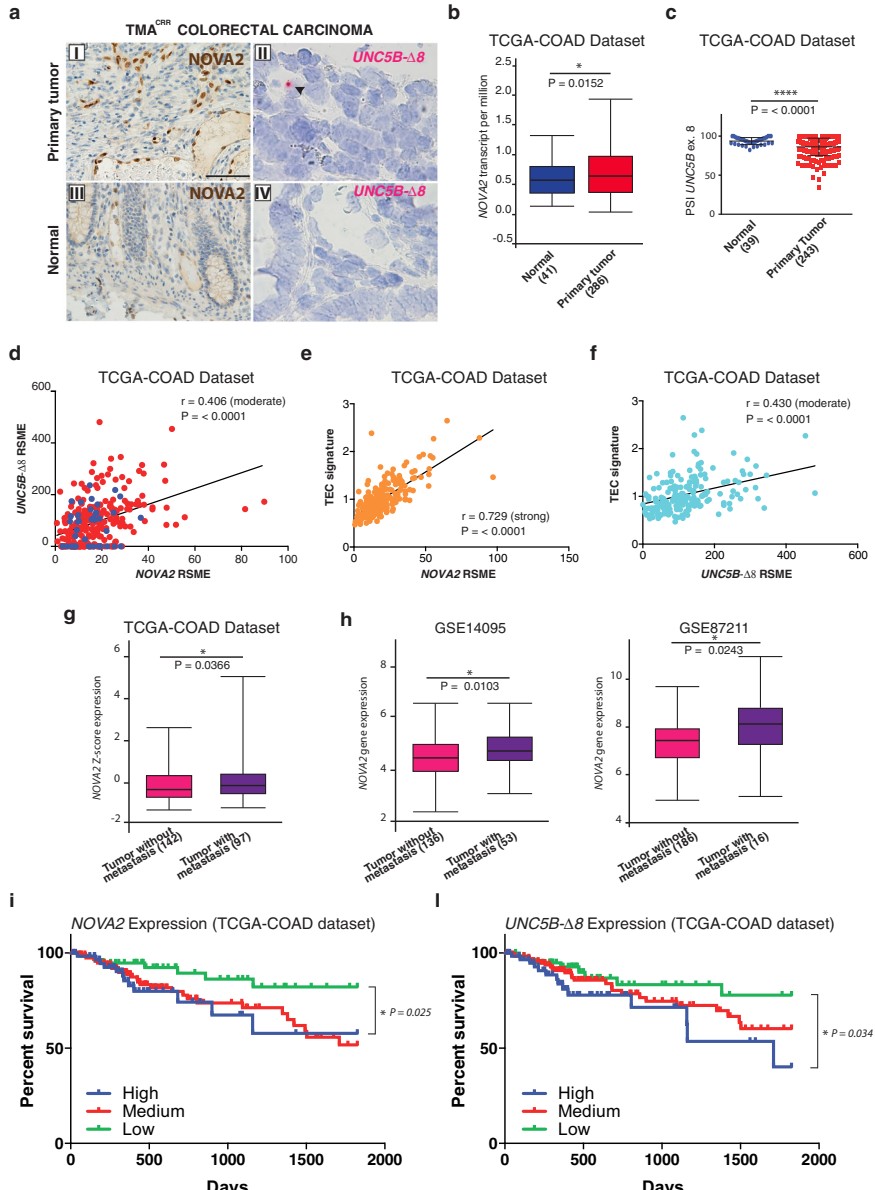

**Fig. 5 *NOVA2* and *UNC5B-Δ8* expression levels in colon cancer patients. a** IHC of NOVA2 (I and III, scale bar: 60 µm) and ISH with the UNC5B-Δ8 probe (II and IV, scale bar: 30 µm) in colon primary tumors and normal adjacent tissues. Black arrowheads indicate *UNC5B-Δ8* signal in ECs in tumor blood vessels. **b** RNA-seq data analysis of *NOVA2* mRNA expression levels, **c** PSI (Percent Spliced-In) of *UNC5B* exon 8, and **d** correlation between *NOVA2* and *UNC5B-Δ8* expression in normal (blue) and colon adenocarcinoma (red) samples in the TCGA-COAD dataset. **e** *NOVA2* and **f** *UNC5B-Δ8* correlation with a colon tumor-specific angiogenetic signature (TEC). **g** *NOVA2* mRNA Z-score expression levels in colon adenocarcinoma without (N0) or with metastasis (N1 or N2) annotated in the TCGA-COAD dataset. **h** *NOVA2* mRNA expression levels in colon adenocarcinoma without or with metastasis (GSE14095 and GSE87211 datasets; http://hcmdb.i-sanger.com). **i** Kaplan–Meier plot of 5-years overall survival in colon adenocarcinoma patients classified according to *NOVA2* or **l** *UNC5B-Δ8* mRNA expression. Blue curve: high expression (1st quartile); red curve: moderate expression; green curve: low expression (3rd quartile). Log-ranked *P*-values are indicated. Boxplot represents upper quartile, median, lower quartile. Error bars indicate minimum and maximum. Two-tailed Log-ranked test (in **b**) or Two-tailed Student's *t*-test (in **c** and **g**) and exact *P* values are indicated: **P* < 0.05, *****P* < 0.0001. *P* values in **h**, were calculated by HCMDB platform. For correlation analysis, linear regression (black line) and Pearson r coefficient with two-tailed *P*-value were also calculated.

a key determinant during capillary morphogenesis and remodeling[13,14,65,66], and its inhibition reduces angiogenesis in vivo[14,65]. In particular, a role for apoptosis in controlling EC number, capillary diameter during angiogenic vessel growth, and maturation has been reported[55]. Moreover, EC apoptosis is instrumental in lumen formation during in vitro vessel-like structure formation[52] and in vivo, where apoptotic ECs have been found during placenta vasculogenesis and angiogenesis processes[53]. While the molecular machinery executing EC

apoptosis is relatively well known, much remains to be understood about how these pathways are regulated during physiological and pathological angiogenesis.

Here we describe an alternative splicing variant of the dependence receptor UNC5B (UNC5B-Δ8) that is produced in ECs, is insensitive to the prosurvival activity mediated by Netrin-1, and regulates vessel formation in an apoptosis-dependent and vascular beds-specific manner. These findings further support the notion that EC apoptosis plays an important role during blood vessel morphogenesis[67–69].

In addition to its known role in axon repulsion and vascular branching[3,4], UNC5B acts as a dependence receptor, as it either prevents or induces apoptosis, depending on Netrin-1 binding[70]. Consequently, the UNC5B/Netrin-1 binding is intimately associated with activation of a cell survival signaling that finely tunes developmental angiogenesis[7].

We demonstrated that *UNC5B* is regulated through AS in ECs, which is mediated by the splicing factor NOVA2 playing relevant roles during angiogenesis and vascular differentiation[27,30]. Mechanistically, we demonstrated that high NOVA2 expression levels promote skipping of *UNC5B* exon 8 by directly binding to a YCAY cluster located in intron 7. This splicing event removes 33 nucleotides encoding for a region predicted to be intrinsically disordered (IDR), located in the extracellular domain proximal to UNC5B transmembrane domain (Supplementary Fig. 16a-b). Interestingly, 90% of AS events located in coding regions have no homology with either functionally defined protein domains or regions with a defined tertiary structure[71,72]. Moreover, AS events of exons that encode for IDRs are frequently regulated in a tissue-specific manner[73,74]. IDRs are commonly found in transmembrane proteins, where they contribute to proper receptor organization, kinase activity, and activation of phosphatase signaling cascades[75]. In addition to providing binding sites for other proteins, nucleic acids, or small molecule ligands[74], IDRs flexibility can favor conformational heterogeneity between structured domains or harbor sites for post-translational modifications (PTM), which ultimately alter downstream signaling through recruitment of different effectors[74]. A salient example is provided by the IDR linking two α-helices (α1-α2) of the antiapoptotic protein Bcl-xL[76], which acts as a flexible loop and regulates apoptosis by allosteric displacement of proapoptotic effectors according to its PTM status[76]. Phosphorylation and deamination of residues within the α1-α2 IDR promote its association with the Bcl-xL core that in this closed conformation is not able to associate with proapoptotic BH3 domain-containing proteins (such as BIM) and p53[76]. We found that the lack of the IDR in UNC5B-Δ8 does not affect its ability to correctly localize, bind Netrin-1, or dimerize. Nevertheless, UNC5B-Δ8 is insensitive to Netrin-1 prosurvival signal. In analogy to the IDR of Bcl-xL, we hypothesize that the absence of the flexibility provided by *UNC5B* exon 8-encoded IDR could alter the cytoplasmic conformation of UNC5B-Δ8 preventing the transduction of Netrin-1 signaling to the cytoplasmic effector proteins (such as DAPk). Accordingly, in UNC5B-Δ8 expressing cells, we found that the levels of the inactive and phosphorylated form of DAPk[77] does not respond to Netrin-1 treatment.

The crystal structure of the UNC5B cytoplasmic domain has provided fundamental cues for the elucidation of UNC5B-mediated signaling during apoptosis[5]. In particular, it has been proposed that the ZU5, UPA, and DD domains can adopt two possible conformations: in the close conformation, ZU5 is associated with both UPA and DD thus suppressing UNC5B capability to promote apoptosis, whereas release of this closed conformation leads to the activation of apoptosis and blood vessel patterning[5]. However, how Netrin-induced receptor dimerization affects the intramolecular changes of these three distinctly folded domains remains unsolved. In addition, we cannot rule out the possibility that the IDR in UNC5B is involved in the formation of a protein complex with other receptors or co-receptors that in turn may impact Netrin-1 or additional UNC5B downstream signaling. Thus, further studies are required to elucidate the role of UNC5B IDR in regulating apoptosis.

In addition to *UNC5B*, NOVA2 regulates AS of pre-mRNAs encoding for other Netrin-1 dependence receptors, such as *DCC* and *Neogenin*[78,79]. In the case of *DCC*, NOVA2 promotes the inclusion of an exon encoding the linker region between the two fibronectin type III domains (FN4 and FN5) recognized by Netrin-1, thus allowing an architectural switch in Netrin-1/DCC receptor assembly[78,80]. Conversely, in *Neogenin* mRNA the exon regulated by NOVA2 encodes for an IDR with unknown function in the cytoplasmic tail of the receptor (Supplementary Fig. 17a). Taken together, these observations suggest a widespread interplay between NOVA2 and the dependence receptor family going beyond their role in axon pathfinding in the central nervous system and potentially affecting other tissues and developmental processes where NOVA2 and dependence receptors are expressed.

*UNC5B* and *NOVA2* genes are highly conserved among different species, including zebrafish[48,50]. In zebrafish, AS of *unc5b*, the IDR encoded by exon 8, and the position of the Nova2 binding sites are also conserved. These observations allowed us to use zebrafish as a model to assess the role of Nova2-mediated AS regulation of *unc5b* in vivo. Several groups have reported PAV formation defects in *unc5b* zebrafish morphants[4,5,51]. Intriguingly, we found that *unc5b-Δ8* is more efficient in rescuing PAV formation defects compared to *unc5b-FL* mRNA. Importantly, we show that EC apoptosis occurs during zebrafish development and is regulated by Unc5b. In particular, Unc5b-Δ8 induces ECs apoptosis in vivo, a biological process required for PAV formation, as demonstrated by the abolishment of *unc5b-Δ8*-mediated PAV rescue in *unc5b* morphants treated with several apoptosis inhibitors. These findings are in line with the observation that a rat UNC5B mutant, which adopts a constitutively open and proapoptotic conformation achieved by deletion of the ZU5 domain, is more potent than UNC5B-FL in restoring PAV formation defects of *unc5b* morphants[5]. Notably, the ability of rat *Unc5b-FL* mRNA to restore PAV formation when injected at high doses, as demonstrated by Wang and colleagues[5], suggests that modulation of *Unc5b* expression has a direct role on UNC5B downstream signaling, possibly by overcoming Netrin-1 prosurvival signal. Our findings also suggest that the forced expression of Unc5b-Δ8, when the endogenous Unc5b is present, is detrimental to the formation of a functional vascular network, thus pointing out the importance of tight regulation of Unc5b-Δ8 expression for the proper development of different vascular structures.

Here, we demonstrated that AS, by generating a Netrin-1 insensitive isoform, provides an additional tissue-specific layer of UNC5B activity regulation bypassing Netrin-1 apoptotic inhibition.

Thus, the ability of UNC5B to regulate angiogenesis and its apoptotic activity are intimately connected during vascular development. However, it is important to note that the effects of UNC5B reduction, differently from the depletion of pleiotropic angiogenic regulators such as VEGFR2[81], occurs in a restricted subset of ECs[51] suggesting that UNC5B function(s), as well as *UNC5B* AS-mediated regulation, could be required in a restricted temporal and spatial manner. Furthermore, other apoptotic-independent UNC5B-DAPk downstream signaling may be differentially affected by the presence of exon 8 IDR, thus conferring an additional layer of complexity to the molecular programs regulating EC functions. Similarly, even if in our in vitro assays we did not observe a difference in protein stability between UNC5B-FL and UNC5B-Δ8, we cannot rule out the possibility that, in specific vascular beds, the absence of exon 8 could alter UNC5B-Δ8 protein half-live thus resulting in a more pronounced proapoptotic activity. Notably, since UNC5B-FL has a partial rescue activity—at least at high doses—and UNC5B-Δ8 also partially rescues PAV formation, it is possible that a certain ratio between UNC5B-FL and UNC5B-Δ8 is required to achieve an optimal proangiogenic activity. This ratio may be different depending on the vessel type or localization, in line with our observation that overexpressing UNC5B-Δ8 interferes with ISV formation.

During cancer progression, the formation of metastasis requires a multistep cascade involving tumor cells' invasion of the surrounding tissues, their intravasation and survival in the circulatory system, and finally their extravasation with the colonization of distant tissues. One centrally important process enabling these steps is the tumor endothelium permeability, which for example can be achieved by tumor ECs lacking intact cell-to-cell contacts[82–84]. Intriguingly, several findings show that EC death is another key process sustaining tumor endothelium fragility and, as a consequence, promoting cancer progression and metastasis formation[85,86]. Indeed, different mechanisms are exploited by solid tumors to induce contact-dependent EC death that in turn contributes to tumor cells escape in the blood circulation and colonization of distant organs[15,16,87–89]. However, in addition to providing the physical space for tumor cells to move across tissues or during vascular co-option[90], ECs apoptosis has been linked to the promotion of human endothelial progenitors differentiation[91] and initiation of EC sprouting[54], thus suggesting a widespread role in signaling or initiating angiogenic EC responses.

Aberrant AS is a hallmark of cancer[92]. Several SRFs act as bona fide oncoproteins[93] since their aberrant expression in tumor cells could drive the production of AS isoforms involved in every aspect of cancer cell biology, such as tumor establishment, progression, and resistance to therapeutic drugs[26,92,94]. Notably, splicing aberrancies are not limited to the cancer cells but are also evident in cells of the tumor microenvironment[95]. Remarkably, an elevated number of AS errors are cancer restricted and particularly relevant for diagnosis, prognosis, and possibly to develop more specific anticancer therapy[94,96].

Here, we found that UNC5B-Δ8 and NOVA2 expression levels are upregulated in ECs of the colon cancer vasculature, are positively correlated to a colon-specific angiogenic signature[62], and are associated with a poor patient outcome. High NOVA2 expression is also associated with shorter metastasis and relapse-free survival. Hence, it is tempting to speculate that the NOVA2-driven UNC5B-Δ8 production restricted to tumor ECs, may represents a post-transcriptional mechanism favoring cancer cells dissemination.

Collectively, our results demonstrated the existence of a post-transcriptional pathway relevant for blood vessel formation. Furthermore, our preliminary and correlative observations suggest a possible role of the UNC5B-Δ8/NOVA2 axis also in tumor angiogenesis. However the complete elucidation of UNC5B-Δ8 and apoptosis roles in the tumor vasculature context requires further studies.

Nevertheless, our findings provide a perspective to understand the molecular underpinnings sustaining the phenotypic and functional aberrancies characteristic of tumor blood vessels, which could be leveraged to develop innovative prognostic markers or more proficient antiangiogenic therapies.

## Methods

**Cell culture**. Mouse embryonic EC (moEC), previously referred to as vascular endothelial (VE) cadherin-positive ECs, were described in[97]. Mouse lung-derived luEC and lu2EC were described in refs.[98] and[99], respectively.

Culture medium of moEC and lu2EC was DMEM (w/glucose 4.5 g/l w/o L-Glutamine; Lonza, #LOBE12614F) supplemented with 10% fetal bovine serum (FBS; Euroclone, #ECS0180L), L-Glutamine (2 mM, Euroclone, #ECB3000D), penicillin/streptomycin (100 U/l, Euroclone, #ECB3001D), HEPES (25 mM, Euroclone, #ECM0180), heparin (100 μg ml/l, from porcine intestinal mucosa; Sigma–Aldrich #H3149), and EC growth supplement (ECGS; 5 μg/ml, Sigma–Aldrich, E2759). The same medium, supplemented with puromycin (3 μg/ml; InvivoGen, #ant-pr-1) or Hygromycin (50 ng/ml; Santa Cruz Biotechnology, #sc-29067), was used for stable moEC depleted or overexpressing NOVA2[27] or carrying inducible Unc5b-FL-GFP, Unc5b-Δ8-GFP, or GFP cDNAs.

LuEC were cultured in MCDB131 medium (Sigma–Aldrich; #M8537) supplemented with 20% FBS, L-Glutamine (2 mM), sodium pyruvate (1 mM),

heparin (100 μg/ml) and ECGS (5 μg/ml). moEC, lu2EC, and HUVEC/TERT2 were grown as sparse or confluent by placing 500,000 cells in 100 mm and 35 mm Petri dishes, respectively. Since NOVA2 mRNA and protein levels are regulated by EC density[27], for the analysis of UNC5B splicing, NOVA2-depleted ECs were used as confluent monolayers, whereas overexpressing HA-tagged NOVA2 ECs were grown as sparse.

Primary human umbilical vein endothelial cells (HUVECs) were obtained from either Yale University Vascular Biology and Therapeutics Core Facility or purchased (Lonza, #CC-2519) and maintained in EGM2-BulletKit medium (Lonza, #CC-3156 and #CC-4176). The cells were verified through CD31 and VE-cadherin staining for EC identity.

Immortalized HUVEC (HUVEC/TERT2) were purchased from Evercyte and grown according to supplier instructions in EBM basal medium (Lonza, #CC-3156) with selected supplements from EGM SingleQuot Kit (Lonza, #CC-4133) (BBE, HEGF, hydrocortisone solution, and ascorbic acid solution), with 10% FBS, and Geneticin (20 μg/ml, Gibco; #10131-019).

Primary cardiac ECs were purified from murine hearts as described in ref.[100] and maintained in EGM2-BulletKit medium (Lonza, #CC-3156 and #CC-4176).

To favor ECs attachment on plastic surfaces, dishes and flasks were coated with Gelatin (0.1% in PBS, Difco).

Culture medium of human cervix carcinoma (HeLa) cells (ATCC, CCL-2), human embryonic kidney (HEK) 293 T cells, and HEK-293A cells (ATCC, CRL-1573) was DMEM high-glucose (w/ 4.5 g/l w/o L-Glutamine) with 10% FBS, L-Glutamine (4 mM), penicillin/streptomycin (100 U/l). NIH/3T3 mouse fibroblast cells were grown in DMEM glucose (w/ 4.5 g/l w/o L-Glutamine) with 10% FBS, L-Glutamine (2 mM), penicillin/streptomycin (100 U/l).

Secreted, recombinant UNC5B-FL and UNC5B-Δ8 ectodomain fragments were produced using FreeStyle™ HEK-293F cells (Life Technologies) cultured in suspension using FreeStyle™ medium (Life Technologies).

All cells were free of mycoplasma contamination. Cells were maintained in a humidified, 5% $CO_2$ atmosphere at 37 °C.

**Zebrafish strains and maintenance**. Zebrafish (*Danio rerio*) experiments were carried out in the zebrafish facilities of the FIRC Institute of Molecular Oncology (IFOM) and University of Milan. Zebrafish from wild-type AB, transgenic Tg(kdrl:GFP)[la116] [101], Tg(fli1a:GFP)[y1] [102], and Tg(kdrl:GFP)[la116;nova2io011] [27] strains were maintained and bred according to standard procedures and national guidelines (Italian decree "4 March 2014, n.26"). All experiments were approved by the Italian Ministry of Health and were performed under the supervision of the institutional organism for animal welfare (Cogentech OPBA). Embryos were collected by natural spawning, staged according to Kimmel and colleagues, and raised at 28 °C in fish water (Instant Ocean, 0.1% Methylene Blue) in Petri dishes, according to established techniques. Generation of *nova2* CRISPR/Cas9 x Tg(kdrl:GFP)[la116] mutants (nova2io011) was described in ref.[27].

**Plasmids**. Mouse Unc5b-FL, amplified by using primers Unc5b-For-HindIII and Unc5b-Rev-EcoRI-GFP and moEC cDNA, was cloned in pEGFP-N1 vector (Clonentech) to add an in-frame C-terminal GFP-tag. Mouse Unc5b-Δ8 was generated by PCR-mediated mutagenesis of Unc5b-FL by using primers Unc5b-deletion_E8-F and Unc5b-deletion_E8-R. Primers Unc5b-For-HindIII and Unc5b-rev-EcoRI- HA were used to generate C-terminal HA-tagged Unc5b-FL and Unc5b-Δ8 cDNAs cloned in the pcDNA3.1(−) vector (Invitrogen; # V79520).

Gateway technology (Gateway LR clonase II, Invitrogen; #11791) was used to generate lentiviral vectors expressing the Unc5b cDNAs under the control of a Tet-on inducible promoter. To this purpose, Unc5b-FL-GFP, Unc5b-Δ8-GFP, or GFP cDNAs were cloned into the pEN-Tmcs vector[103] and then transferred in the pSLIK-Hygro plasmid (Addgene, #25737)[103]. The same technology was used to clone in the adenoviral construct pAd/CMV/V5/DEST (Invitrogen; #11791) the GFP-tagged Unc5b cDNAs, which were previously subcloned into the pENTR11 vector (Invitrogen; #A10467).

To generate Unc5b wild-type minigene, the Unc5b genomic region encompassing exons 7 and 9 was PCR amplified from mouse fibroblast DNA with primers Unc5b_minigene_F and Unc5b_minigene_R and cloned into the pcDNA3.1 (+) vector (Thermo Fisher, #V790-20). Mutated Unc5b minigene was generated by site-directed mutagenesis (Phusion Site-Directed Mutagenesis Kit, Thermo Fisher, #F541) with primers Unc5b_minigene_mut_F and Unc5b_minigene_mut_R.

Morpholino-resistant zebrafish unc5b-FL and unc5b-Δ8 cDNAs, amplified with primers Unc5b_zebrafish-F/Unc5b_zebrafish-R and an RT reaction generated with RNA extracted from Tg(Kdrl:GFP)[la116] embryos at 72 hpf, were cloned in a modified pCS2+ (RZDP) vector in-frame C-terminal mCherry tag. The plasmid expressing DAPk and L1CAM were described in[75] and[28], respectively.

For secreted recombinant protein production using HEK-293F cells we used the pHLSec-UNC5B-FL vector[56], kindly provided by Prof. Elena Seiradake, and generated the corresponding pHLSec-UNC5B-Δ8 through site-directed mutagenesis, using primers Unc5bEcto-deletion_E8-F and Unc5bEcto-deletion_E8-R. Primers used for cloning experiments, RT-PCR, and RT-qPCR are listed in Supplementary Table 1. All constructs were verified by sequencing.

**Lentivirus and adenovirus production and transduction**. For lentivirus production HEK-293T cells were seeded in DMEM supplemented with 10% FBS without antibiotics in 60 mm Petri dishes. The day after, cells at 60–70% confluence were transfected with 5 µg of packaging plasmid, 5 µg of envelope plasmid, and 20 µg of lentiviral vectors carrying the *Unc5b* cDNAs. After 18 h, the medium was replaced with DMEM supplemented with 20% FBS and 1% penicillin/streptomycin. Cells were incubated for an additional 24 h and the medium containing the lentiviral particles was harvested, filtered using a 0.45 µm filter unit, and used to infect ECs. For viral transduction, moEC were seeded in 100 mm Petri dishes and infected at 70% of confluence. These cells were incubated overnight with the viral supernatant supplemented with 0.2 mM proline and polybrene (final concentration 8 µg/ml; Sigma–Aldrich). After 48 h, hygromycin selection (50 µg/ml) was started and continued until all noninfected control cells died.

Adenovirus production was performed into HEK-293A cells seeded in 100 mm Petri dishes in DMEM medium without antibiotics. After 24 h, HEK-293A (at the 50–70% of confluence) were transfected with the adenoviral constructs carrying the *Unc5b-GFP* cDNAs (previously linearized with PacI) by using Lipofectamine 2000 (Invitrogen; #11668027). On day 3 after seeding, the medium was replaced and, after additional 12 days, cell supernatant was collected and frozen at −80 °C. Upon three cycles of freeze and thaw, medium containing the adenoviral particles was harvested, sterile-filtered, and used to infect HUVECs. Adenoviral infection was performed by incubating ECs with the viral supernatant for 24 h.

**Plasmid transfection**. HeLa and moEC were transiently transfected with Lipofectamine 3000 (Invitrogen; #L3000001), according to the manufacturer's protocol. Briefly, cells were seeded in a 6-well or 12-well plate to reach 70–75% of confluence the day of the transfection.

For recombinant UNC5B ectodomains production, HEK-293F cells were transfected at a confluence of $10^6$ cells/ml, using 1 µg of plasmid DNA and 3 µg of linear polyethyleneimine 25,000 Da (PEI; Polysciences). Four hours after transfection, cell cultures were supplemented with Primatone® RL (Sigma–Aldrich) at a final concentration of 0.6%.

**siRNA-mediated RNA interference**. RNA interference was carried out on ECs using Lipofectamine RNAiMAX (Invitrogen; #13778030) following the supplier's protocol. Briefly, HUVEC/TERT2, luEC, and lu2EC were transfected at 80–90% confluence with the indicated siRNAs. HUVEC/TERT2 were silenced with a siRNA against the human *NOVA2* gene (MISSION siRNA ID SASI_Hs01_00220812; Sigma–Aldrich), mouse *Nova2* gene (MISSION siRNA ID SASI_Mm01_0094763; Sigma–Aldrich) human *UNC5B-Δ8* transcript (CUGUGCAUGCAAAUGCUG-GAdTdT/UCCAGCAUUUGCAUGCACAGdTdT), or with a control siRNA (MISSION siRNA Universal Negative Control #1; Sigma–Aldrich). Two subsequent transfections (with 24 h intervals) were performed with 30 nM of each siRNA. Cells were collected 48 h after the second transfection. To obtain HUVECs stably expressing shRNA-NOVA2, we produce lentiviral vectors in HEK-293T, as above. Lentiviruses expressing a scramble sequence (*shCtr*) were used as control. At 48 h after transduction, HUVECs were selected using puromycin (1 µg/ml).

**Morpholino treatment of human ECs**. Confluent (90%) HUVEC/TERT2 were transfected with MO-i7_UNC5B oligonucleotide (Gene Tools) at 15 µM plus 6 µM Endo-Porter PEG system (Gene Tools) according to the manufacturer's instructions. A standard control oligonucleotide (Gene Tools) was used as control. After 36 h cells were detached and seeded in Matrigel-coated plates for tube formation capability evaluation. UNC5B AS was evaluated after 48 h of morpholino treatment. Morpholino oligo sequences are listed in Supplementary Table 2.

**In vitro tube formation assay**. To assess the ability of ECs to organize in a capillary-like network, 30,000 HUVEC/TERT2 cells were plated in complete medium on a 48-well plate coated with 100 µl/well of Growth Factor-Reduced Matrigel (BD Biosciences, #356231). Tube-like structures were manually counted under the microscope at the indicated timepoints. Tube-like structure formation of UNC5B-Δ8-depleted HUVEC/TERT2 was evaluated in 96-well plate coated with 50 µl/well of Matrigel in complete medium at least 24 h after the last siRNA transfection or 36 h after morpholino treatment.

**Cell imaging**. Transiently transfected HeLa cells were fixed with 4% paraformaldehyde (PFA; Sigma–Aldrich). Nuclei were stained with 0.1 g/ml DAPI (Sigma–Aldrich). The same protocol was also used for transduced ECs (moEC and HUVEC), which were seeded in 35 mm Petri dishes coated with 0.1% Gelatin. For imaging, an epifluorescence microscope (Optical Microscope Olympus IX71) equipped with ×4 or ×60 objectives was used. Photomicrographs were taken with a digital camera Cool SNAPES (Photometrics). Data acquisition was done using the MetaMorph 7.7.5 software (Universal Imaging Corporation). Images were exported to Photoshop (Adobe). No manipulations were performed other than adjustments in brightness and contrast. High-resolution pictures were acquired using a Leica SP5 confocal microscope with a Leica spectral detection system (Leica 15 SP detector) and the Leica application suite advanced fluorescence software or Zeiss LSM800 confocal microscope and Zeiss Zen 2.3 Software.

**RNA extraction, RT-PCR, and RT-qPCR**. Total RNA was isolated using the RNeasy Mini Kit (QIAGEN, #74106) according to the manufacturer's instructions. Total RNA was extracted from matrigel-based tube-like structures by using the ReliaPrep™ RNA Miniprep System (Promega, #Z6011) according to the manufacturer's instructions (#AN296). Total RNA from wild-type AB, *Tg(fli1a:GFP)^{y1}*, *nova2* morpholino-mediated knockdown *Tg(fli1a:GFP)^{y1}* embryos, and *Tg(kdrl: GFP)^{la116;nova2io011}* mutants were extracted with TRIzol reagent (Invitrogen. #15596026) and purified with the RNeasy Mini Kit (QIAGEN, #74106).

Human primary colorectal cancer, adjacent normal colorectal, and liver metastasis snap-frozen tissue samples were obtained from Biological Resource Centre (CRB, Centre de Resource Biologique) of Léon Bérard cancer Centre (Lyon, France) (protocol number: BB-0033-00050), as previously described[104]. CRB medical and scientific committee approved the protocol. All patients signed informed consent according to the French laws. Total RNA was extracted with RNeasy Mini QIAcube Kit (Qiagen, #74116), following the manufacturer's protocol. RNA amount and quality were evaluated by NanoDrop 1000 Spectrophotometer (Thermo Scientific) and TapeStation System (Agilent Technologies), respectively.

After treatment with DNAse I (Ambion, #AM2222), 0.5–1.5 µg of purified RNA was retro-transcribed with a mix of oligos d(T)$_{18}$ and random hexamers by using SuperScript IV First-strand System (Invitrogen, #18091050). Resulting cDNA (1/20 v/v) was then PCR amplified with the GoTaq G2 Flexi DNA Polymerase (Promega, #M7805) according to the manufacturer's instructions. For RT-qPCR, an aliquot of the RT reaction was analyzed with QuantiTect SYBR Green PCR (QIAGEN, #204145) by using LyghtCycler 480 (Roche). Target transcript levels were normalized to those of the indicated reference genes. The expression of each gene was measured in at least three independent experiments. All PCR products were sequenced and bands intensity was quantified with NIH Image J software (version 1.48 v). The percentage of exon inclusion was calculated as the ratio between the intensity of the band with the alternative exon included and the intensity of all bands.

**RNA-binding protein target motifs prediction**. RBP motifs prediction was performed by using RBPmap web server (http://rbpmap.technion.ac.il)[41]. Human and mouse *UNC5B* pre-mRNA sequences at exon 8 splice junctions, extending 200 nucleotides into introns and 50 nucleotides into exons, were analyzed for human/mouse motifs. The same pre-mRNA regions were analyzed by using SpliceAid 2 (http://www.introni.it/spliceaid.html)[42].

**Minigene splicing assay**. *Unc5b* minigenes (wild-type or mutated) were co-transfected into moEC with protein expression vectors (HA-NOVA2 or T7-HNRNPA1, previously generated in our laboratory[27,103,105] or empty vector at 1:1 ratio by using Lipofectamine 3000. After 24 h, total RNA and protein lysates were collected.

**In vitro RNA pull-down assay**. RNA probes containing the identified cluster of NOVA2 binding sites in mouse *Unc5b intron 7* or the same region in which YCAY sites were substituted with YAAY tetranucleotides to prevent NOVA2 binding were generated with the following set of primers: T7_YCAYcluster_F/ TDP43_YCAYcluster _R.

Sense and antisense oligos carried a T7 polymerase promoter sequence and a consensus-binding motif for TDP-43, respectively. cDNA template for in vitro transcription was generated with Platinum SuperFi Green DNA polymerase (Thermo Fischer Scientific, #12357-010) and used (1 µg) for in vitro synthesis of RNA transcripts using the MAXIscript T7 Transcription Kit Life (Technologies, #AM1312) according to the manufacturer's protocol. Unincorporated nucleotides were removed through two successive ammonium acetate/ethanol precipitations. Next, 40 pmol of unlabeled RNA were biotinylated at the 3' terminus using the Pierce RNA 3' End Biotinylation kit (Thermo Fisher Scientific, #20160) following the manufacturer's instructions.

For in vitro pull-down assays biotinylated RNA probes were resuspended in 250 µl of 2× TENT buffer (20 mM Tris-HCl pH 8; 2 mM EDTA; 500 mM NaCl; 1% Triton X-100), protease inhibitors (Roche, cOmplete Mini; #11836170001), RNAse inhibitor (SUPERase In, Ambion, AM2694), and 250 µl of cell lysates of NOVA2-HA overexpressing HeLa cells for 30 min at RT. Seventy-five microliters of high capacity streptavidin agarose resin (Thermo Fischer Scientific, #20357) was added to the mixture and incubated for an additional 30 min at RT with intermittent mixing every 5 min. Beads were then pelleted (2000 × g at 4 °C) and washed four times in 1× TENT buffer. Elution of proteins bound to RNA probes was performed in 2× SDS-sample buffer (120 mM Tris/HCl pH 6.8; 20% glycerol; 4% SDS, 0.04% bromophenol blue; 10% β-mercaptoethanol) and by heating the samples at 95 °C for 10 min. Standard western blotting techniques were used to detect NOVA2-HA and TDP-43 in the pull-down.

**Vital exclusion dye assay**. HeLa cells were transfected and serum-starved (Complete DMEM with 0.2% FBS) after 24 h. Mouse recombinant Netrin-1 (150 ng/ml in PBS/0.2% BSA, Bovin Serum Albumin; R&D) or the equivalent amount of PBS/0.2% BSA were added during starvation. Collected cells were then incubated (5 min) with a vital exclusion dye (Erythrosin B, 0.05% in PBS), which is

impermeable to biological membranes[106], and counted for positive and negative staining. Percentage of cell death was calculated as the number of Erythrosin B-positive cells with respect to the total. Cell death index is the ratio between the percentage of cell death of each sample and the control (GFP untreated). moEC plated at 70% of confluence were induced with Doxycycline (1 µ/ml) in a serum-starved medium (Complete DMEM with 0.2% FBS; w/o ECGS) for 6 or 24 h and then tested with the same procedure described for HeLa cells.

**Caspase-3 activity assay**. Caspase-3 activity was determined using the "Caspase Glo 3/7" luminescence assay (Promega; #G8090) in a Synergytm HT Microplate reader (BioTeK Instruments). Briefly, 10,000 HUVEC were seeded in a 96-well plate and transduced with the indicated adenovirus. The next day, cells were serum-starved (EBM2 medium with 0.2% FBS), treated with mouse recombinant Netrin-1 (150 ng/ml) or the equivalent amount of PBS/0.2% BSA, and assayed 8 h for Caspase-3 activity following manufacturing instructions.

**Co-immunoprecipitation**. Co-immunoprecipitation assays were carried in HeLa cells transiently co-transfected with plasmids expressing UNC5B isoforms or DAPk (described in ref. [56]) and treated with Netrin-1 (150 ng/ml) or PBS/0.2% BSA for 30 min (UNC5B dimerization) or 3 h (DAPk interaction). Adherent cells were washed with ice-cold PBS and collected in an IP Buffer (50 mM HEPES pH 7.6, 150 mM NaCl, 5 mM EDTA, and 0.1% NP-40 with the addition of protease and phosphatase inhibitors). GFP-tagged UNC5B splicing variants were immunoprecipitated using GFP-trap magnetic beads (Chromotek; #gtma) for 1 h at 4 °C. Magnetic beads were washed three times in Wash Buffer (50 mM HEPES pH 7.6, 150 mM NaCl, 5 mM EDTA) and immunoprecipitated proteins were eluted in a 2× SDS-sample buffer (120 mM Tris/Cl pH 6.8; 20% glycerol; 4% SDS, 0.04% bromophenol blue; 10% β-mercaptoethanol) by heating at 95 °C for 10 min. Half of the eluted proteins were loaded in an SDS-PAGE gel and analyzed by immunoblotting with the indicated antibodies. As a negative control, GFP-tagged Unc5b-FL were immunoprecipitated with aspecific magnetic beads.

**Protein surface biotinylation**. UNC5B-FL-HA, UNC5B-Δ8-HA, and L1CAM-HA overexpressing HeLa cells were incubated with EZ-Link Sulfo-NHS-SS-Biotin (Life Technologies, #21331) 0.2 mg/ml in chilled CM-PBS at 4 °C for 30 min. After three washes (10 min each) in quenching buffer (100 mM glycine, 2.5 mM CaCl$_2$, 1 mM Mg$_2$Cl$_2$ in PBS) cells were collected by scraping and lysed in RIPA buffer (1% Triton X-100, 1% Sodium deoxycholate, 0.1% SDS, 1 mM EDTA, 1 mM EGTA, 59 mM NaF, 160 mM NaCl, and 20 mM Tris-HCl, pH 7.4) supplemented with protease inhibitors (PI) (cOmplete™ and EDTA-free Protease Inhibitor cocktail; Roche). Biotinylated proteins were retrieved by using Pierce High Capacity Streptavidin agarose beads (Life Technologies, #20359). After three washed in RIPA + PI buffer, biotinylated proteins were eluted in a 2× SDS-sample buffer by heating at 95 °C for 10 min. Eluted proteins were loaded in an SDS-PAGE gel and analyzed by immunoblotting with the indicated antibodies. A small fraction of protein lysate for each condition was saved as an input fraction.

**Immunoblot analysis**. Cells were lysed in Laemmli buffer, supplemented with protease and phosphatase inhibitors (cOmplete™ and EDTA-free Protease Inhibitor cocktail; Roche). Proteins were separated in SDS-PAGE and analyzed by western blotting by standard procedures. After protein transfer, the nitrocellulose membranes (0.45 µm; Whatman PROTRAN) were blocked by incubation with 5% non-fat dry milk. The following primary antibodies were used: anti-NOVA2 C-16 (1:200; Santa Cruz Biotechnology, #sc-10546), anti-α-TUBULIN (1:100,000; Sigma–Aldrich, #T9026), anti-haemagglutinin (HA; 1:1,000; Roche, #11867423001), anti-GAPDH (1:5,000; Abcam, #ab75834), anti-GFP (1:3,000; Millipore, #MAB3580); anti-Cleaved Caspase-3 (1:1,000; Cell Signaling, #9661); anti-VINCULIN (1:5,000; Millipore, #MAB3574); anti-UNC5B (1:1,000; Cell Signaling, #13851), anti-DAPk pSer308 (1:100,000; Sigma–Aldrich, #D4941); anti-DAPk (1:1000; Sigma–Aldrich, #D1319); anti-T7 (1:1,000; Novagen, #69522-3); anti-RBFOX2 (1:2,000; Bethyl Laboratories, A300-864A); anti-mCherry (1:1,000; Chromotek, #5f8); TDP-43 (1:1,000; Proteintech, #10782-2-AP). The following secondary antibodies linked to horseradish peroxidase (Jackson Immuno Research) were used: anti-Mouse (1:5,000; #115-035-146), anti-Goat (1:5,000; #705-035-147), anti-Rat (1:5,000; #112-035-175) and anti-Rabbit (1:10,000; #711-035-152). Immunostained bands were detected using the chemiluminescent method (Euroclone, LiteAblot Plus/Extended, #EMP011005/#EMP013001).

**Morpholino and mRNA injections of zebrafish embryo**. Zebrafish *nova2* morphants were previously described in ref. [27]. For *unc5b* knockdown, zebrafish embryos at one- to two-cell stage were injected with an *unc5b* morpholino antisense oligonucleotide (MO-*unc5b*) designed to block splicing of intron 1 as described by Lu and colleagues[4]. Morpholino efficiency was evaluated by RT-PCR with RNA extracted from 72 hpf embryos and with primers annealing in *unc5b* exon 1 and intron 1. Capped zebrafish *unc5b-FL* and *unc5b-Δ8* mRNAs were transcribed in vitro by using the SP6 mMessage mMachine kit (Ambion; #AM1340) and co-injected with MO-*unc5b* (0.03 pmol/embryo) in one-cell stage zebrafish embryos. *unc5b-Δ8* mRNA was also injected in Tg(*fli1a:GFP*)$^{y1}$ zebrafish embryos following the same experimental procedure. When not differentially indicated 100 pg of mRNA per embryo have been injected. Where indicated, 40 nM BAF (Boc-d-FMK; Axon Medchem, #EC2158), 50 µM Q-VD-OPh hydrate (Adooq Bioscience; #A14915), or 300 µM Z-VAD-FMK (ApexBio Technology; #A1902) were added in the E3 medium 4 h after MO-*unc5b* and *unc5b-Δ8* mRNA co-injection.

Evaluation of vascular defects was performed by immunofluorescence staining of zebrafish embryos. Tg(*kdrl:GFP*)$^{la116}$ and Tg(*fli1a:GFP*)$^{y1}$ zebrafish embryos at the indicated developmental stage were dechorionated and fixed in 4% PFA in PBS, overnight (ON) at 4 °C. Embryos were then washed four times for 5 min in PBST (PBS, 0.1% Tween 20). Permeabilization in PSBT with 0.5% Triton X-100 was performed for 30 min at room temperature (RT). Embryos were then blocked for 2 h at RT in a solution of PBST plus 0.5% Triton X-100, 10% normal goat serum and 1% BSA. Embryos were then incubated with primary antibodies in blocking solution ON at 4 °C. Successively, embryos were washed six times in PBST over 4 h at RT and then incubated with secondary antibodies in blocking solution, ON at 4 °C. Embryos were washed six times in PBST over 4 h at RT and equilibrated in glycerol 85% in PBS. The following antibodies were used: anti-GFP (1:2,000; Millipore, #MAB3580); anti-mouse Alexa-488-conjugated IgG (1:400; Thermo Fischer Scientific, #A-32766); anti-cleaved Caspase-3 (1:200; Cell Signaling, #9661). For the microscope analysis, we mounted on slides the trunk and tail regions dissected from five to six embryos of each samples. Images were taken with a Leica TCS SP2 confocal microscope, using oil-immersion objective. The same mounting procedures and magnifications were used when comparing the different experimental groups of each experiment. At least 20 embryos of each sample were dissected to mount on slide trunk and tail regions.

To rescue vascular defects of *nova2* mutants, one- to two-cell stage embryos from heterozygous *nova2* mutants were injected with in vitro transcribed *unc5b-FL* and *unc5b-Δ8* mRNAs. At 72 hpf, embryos were analyzed by immunofluorescence staining; for each biological experiment at least 20 embryos of each sample were dissected to mount on slide trunk and tail regions.

**Terminal deoxynucleotidyl transferase (TdT) dUTP Nick-End Labeling (TUNEL) assay**. EC death in Tg(*kdrl:GFP*)$^{la116}$ and Tg(*fli1a:GFP*)$^{y1}$ embryos at different developmental stages was evaluated by using the Invitrogen™ Click-iT™ Plus TUNEL Assay for In Situ Apoptosis Detection, Alexa Fluor™ 647 dye (Invitrogen; #C10619) following manufacturer's instructions. TUNEL staining was validated in 24 hpf Tg(*fli1a:GFP*)$^{y1}$ zebrafish embryos treated with DNAse (provided in the kit). Briefly, embryos were fixed in 2% PFA and dehydrated in absolute methanol ON at −20 °C. Embryos were then rehydrated, washed with PBST and permeabilized for 30 min with proteinase-K. Embryos were then fixed in 2% PFA 20 min, washed in PBST and treated 10 min with TdT reaction buffer. After incubation with TdT reaction cocktail for 1 h at 37 °C embryos were washed in 3% BSA in PBST and incubated in Click-IT reaction cocktail 30 min at RT. Embryos were finally washed six times in PBST over 4 h, equilibrated in glycerol 85% in PBS and mounted for confocal acquisition. For GFP detection, anti-mouse Alexa-488-conjugated IgG (1:400, Thermo Fischer Scientific, #A-32766) was used following manufacturers instructions.

**Crosslinking and immunoprecipitation (CLIP)**. CLIP assay was performed as previously described[28,107,108,109]. Briefly, moEC were irradiated with 150 mJ/cm$^2$ in a Stratlinker 2400 at 254 nm. After centrifugation at $1,300 \times g$ for 3 min, cell pellet was incubated for 10 min on ice with lysis buffer [50 mM Tris-HCl, pH 7.4, 100 mM NaCl, 1% Igepal CA-630 (Sigma–Aldrich), 0.1% SDS, 0.5% sodium deoxycholate, 0.5 mM Na$_3$VO$_4$, 1 mM DTT, protease inhibitor cocktail (Sigma–Aldrich), and RNase inhibitor (Promega)]. Samples were briefly sonicated and incubated with 10 µl of 1:1,000 RNase I (100 U/µl, Ambion) dilution and 2 µl of DNase (2 U/µl, Ambion) for 3 min at 37 °C shaking at $100 \times g$, and then centrifuged at $15,000 \times g$ for 10 min at 4 °C. The amount of 1 mg of extract was immunoprecipitated with anti-NOVA2 C-16 (1:200; Santa Cruz Biotechnology, #sc-10546) antibody or purified IgG (negative control) in the presence of protein A/G magnetic Dynabeads (Life Technologies). Immunoprecipitates were incubated ON at 4 °C under constant rotation. After stringent washes with high salt buffer (50 mM Tris-HCl, pH 7.4, 1 M NaCl, 1 mM EDTA, 1% Igepal CA-630, 0.1% SDS, 0.5% sodium deoxycholate), beads were equilibrated in PK buffer (100 mM Tris-HCl, pH 7.4, 50 mM NaCl, 10 mM EDTA). An aliquot (10%) was kept as input lysates, while the rest was treated with 50 µg Proteinase-K and incubated for 20 min at 37 °C shaking at $100 \times g$. 7 M urea was added to the PK buffer and incubation was performed for further 20 min at 37 °C and $100 \times g$. The solution was collected and phenol/CHCl$_3$ (Ambion) was added. After incubation for 5 min at 30 °C shaking at $100 \times g$, phases were separated by spinning for 5 min at $17,000 \times g$ at RT. The aqueous layer was transferred into a new tube and precipitated by the addition of 0.5 µl glycoblue (Ambion), 3 M sodium acetate pH 5.5, and 100% ethanol. After mixing, the solution containing retained RNA was precipitated ON at −20 °C. RNA extracted from both the input material and the immunoprecipitates was then analyzed by RT-qPCR as described in the above paragraph. The binding was expressed as a percentage of the input material.

**Production of the UNC5B-FL and UNC5B-Δ8 ectodomain recombinant protein**. Suspension-cultured HEK-293F cells transfected using either pHLSec-

UNC5B-FL or pHLSec-UNC5B-Δ8 plasmids were harvested 6 days after transfection by 10 min centrifugation at $1,000 \times g$[110]. The supernatant containing secreted UNC5B variants was loaded at 0.5 ml/min on a 1 ml HisTrap Excel 5 mL column (GE Healthcare). The column was washed with 10 column volumes of buffer A, composed of 50 mM HEPES, 500 mM NaCl, pH 8.0. Nonspecifically bound material was washed out by supplementing 35 mM imidazole to buffer A. Elution of the sample of interest was obtained by increasing the imidazole concentration to 250 mM. The elution peak protein was concentrated by centrifugation using Vivaspin® Turbo 15, 30,000 MWCO centrifugal filters (Sartorius) to a volume less than 500 µl and then loaded onto a Superdex 200 10/300 GL (GE Healthcare) gel filtration column equilibrated with 50 mM HEPES, 200 mM NaCl, pH 8.0. Both samples yielded single peak elutions corresponding to the UNC5B variants as assessed by SDS-PAGE analysis. Peak fractions were collected and further concentrated to 1 mg/ml. The purified samples were then flash-frozen in liquid nitrogen and stored at −80 °C until further usage.

**Surface Plasmon Resonance.** Surface Plasmon Resonance analyses were carried out on a BIAcore T200 (GE Healthcare) instrument at 25 °C. Netrin-1 (Netrin-1-Fc; ProSci, #90-261) was immobilized on a Protein G sensor chip (GE Healthcare) through injection at a concentration of 30 µg/ml. Recombinant UNC5B-FL and UNC5B-Δ8 ectodomains analytes, prepared in PBS, 0.005% Tween 20, were injected at different concentrations (0–10 µM) using a 30 µl/min flow rate. Proteins were allowed to associate for 120 sec and to dissociate for 600 sec. Data were processed with the BIAcore BIAevaluation software v. 3.0.

**Tissues and immunohistochemistry.** Tissues sections of tissue microarray (TMA^CCR) slides were obtained from a set of colorectal carcinomas ($n = 67$) and included cases with ($n = 33$) or without ($n = 34$) KRAS mutations on exon 12 or 13. Clinical samples were represented by a multitumor TMA (including cores from 126 primary carcinomas) and paraffin tissue blocks from archival colorectal carcinomas were retrieved from the archive of Pathology (U.O. Anatomia Patologica, Spedali Civili di Brescia). The study supervised by W.V., was approved by the local IRB (code: WW-IMMUNOCANCERhum, NP-906). Full sections from carcinoma of the oral cavity ($n = 5$) and hepatocellular carcinoma ($n = 5$) were included in the analysis. Sections of 4 µm were obtained for immunohistochemistry by using anti-NOVA2 (polyclonal rabbit, 1:100; Sigma–Aldrich, #HPA045607). Antigen retrieval was performed using EDTA Buffer (pH 8.0) whereas signal detection was obtained using Novolink Polymer System (Novocastra) followed by DAB. Anti NOVA2 was combined with anti-CD31 (clone PECAM-1, 1:100, Leica; #CD31-PECAM-1) and anti-Cleaved Caspase-3 (polyclonal rabbit, 1:600, R&D Systems; #AF835). For double staining, after completing the first immune reaction, the second was visualized using Mach 4 MR-AP (BIOCARE MEDICAL), followed by Ferangi Blue (BIOCARE MEDICAL). For triple immunohistochemistry, the third immune reaction was revealed using Mach 4 MR-AP (Biocare Medical), followed by ImmPACT Vector Red Substrate Kit, Alkaline Phosphatase (VECTOR LABORATORIES).

Validation of Cleaved Caspase-3 antibody used for triple immunohistochemistry analysis was performed in Burkitt's Lymphoma, a tumor-cell population know to display high rates of apoptosis in vivo[109,107], and Reactive lymph nodes, as control. Isotype and omission controls on colorectal carcinomas sections are also shown.

**Digital microscopy.** Stained slides were acquired using the Aperio CS2 digital scanner and ScanScope software (Leica Biosystems, Wetzlar, Germany). Images were viewed and organized using ImageScope software (Leica biosystems, Wetzlar, Germany). Region of interest consisted of the tumor area and normal colon; necrotic areas were excluded from the selection. IHC Nuclear Image Analysis algorithm (Leica biosystems, Wetzlar, Germany) was setup for the analysis to identify categories of strong and weak positive cells based on the signal intensity (Supplementary Fig. 18). Data are expressed as the absolute number of NOVA2-positive cells per mm$^2$ and as fractions of strong and weak positive cells.

**Chromogenic BaseScope RNA in situ hybridization.** A custom-designed specific probe targeting UNC5B-tv2-E7E8 (gene accession NM_001244889, 1452–1491 bp) was obtained by design a 1ZZ probe against human Unc5b exon 7-exon 9 junction; a control probe was represented by Hs-PPIB-3zz probe. All probes and BaseScope reagents were purchased from Advanced Cell Diagnostic (ACD). The reaction was revealed using the BaseScope TM detection Reagents v2-RED (#323910). Cell blocks obtained by NIH/3T3 murine cells transfected with human UNC5B-Δ8 were used as a positive control; negative controls were represented by murine NIH/3T3 murine cells transfected with human UNC5B-FL or empty vector. 4 µm sections obtained from TMA^CCR and control cell blocks were baked in a 60 °C oven for 30 min and deparaffinized in three changes of xylene and three changes of absolute alcohol. The sections were blocked with 3% H$_2$O$_2$ (#322381) for 10 min at RT and washed with deionized water. Target retrieval was performed in a controlled water bath with ACD target retrieval solution (#322000) at 98 °C for 15 min. A permeabilization step was included using protease III (#322380) with incubation in a humidified chamber (30 min for FFPE tissue and 15 min for cell pellets) in a 40 °C oven (HB-1000, Hybridizer, UVP Cambridge, England). After washing with

deionized water the target probe and the housekeeping gene-positive control probe were incubated for 2 h in a humidified chamber in a 40 °C oven (HB-1000 Hybridizer, UVP, Cambridge, England). A subsequent series of amplification steps were performed according to the protocol as follows: Amplification AMP1 (30 min at 40 °C), AMP2 (30 min at 40 °C), AMP3 (15 min at 40 °C), AMP4 (30 min at 40 °C), AMP5 (30 min at 40 °C), AMP6 (15 min at 40 °C) with the humidified chamber and AMP7 (30 min at RT) and AMP8 (15 min at RT). Slides were washed twice for 2 min in between each amplification step using diluted washing buffer (#310091). Chromogenic detection was performed with FAST RED-B (#322919) diluted 1:60 in FAST RED-A buffer (#322918) incubated for 10 min at RT. Nuclear counterstaining was performed with Mayer's hematoxylin. Finally, slides were covered using glycerol.

**Gene expression and splicing analyses of transcriptomic data.** TCGA-COAD, TCGA-LIHC, and TCGA-HNSC cancer omics data were analyzed by using the UALCAN (http://ualcan.path.uab.edu)[59], a web-portal allowing gene expression analysis of selected genes in TCGA level 3 RNA-seq data corresponding to primary tumors and normal samples. Transcripts per million (TPM) were used to generate boxplots. Data are visualized representing interquartile ranges (minimum, 25th percentile, median, 75th percentile, maximum). Outliers are excluded. UALCAN also provides an estimate of the statistically significant difference in gene expression levels between groups by employing TPM values and a t-test.

Gene expression data were also retrieved by using Oncomine (http://oncomine.org/resource/)[61]. Colon adenocarcinoma microarray databases with altered NOVA2 gene expression were filtered considering their significance ($P$ value < 0.05). The following probes are shown in Supplementary Fig. 14a-b: A_23_P101609 (TCGA Colorectal Dataset) and 206476_s_at (Kaiser Dataset). 90th percentile, 75th percentile, median, 25th percentile, 10th percentile are represented.

UNC5B mRNA splicing patterns (inclusion of exon 8) were analyzed by using TCGA SpliceSeq (http://projects.insilico.us.com/TCGASpliceSeq)[111], a web-based resource that provides Percent Spliced-In (PSI) values for splicing events on samples from the TCGA-COAD, TCGA-LIHC, and TCGA-HNSC level 3 datasets.

Metastasis status information, as specified by the neoplasm disease lymph node stage of the American joint committee on cancer code (N0, N1, or N2), of colon adenocarcinoma patients (TCGA, Firehose Legacy) were retrieved by using cBioPortal for Cancer Genomics (https://www.cbioportal.org)[63]. Associated mRNA expression Z-scores (RNA-Seq V2 RSEM) were plotted according to the presence of lymph node metastasis (N1, N1a, N1b, N1c, N2, N2a, and N2b status) or absence (N0). Additional colorectal cancer datasets with metastasis information were retrieved by using the Human Cancer Metastasis Database (HCMDB) (https://hcmdb.i-sanger.com), which directly provides differentially expressed genes between different types of samples[64].

**Correlation analyses.** Correlation analyses were performed by using Pearson's correlation method. The Vertebrate Alternative Splicing and Transcription Database (VastDB; http://vastdb.crg.eu/)[37] was used to retrieve human UNC5B exon 8 PSI values and SRSF1, NOVA2, HNRNPA1, SRSF3, HNRNPC, or RBFOX2 cRPKM expression values in 71 sets of tissues, cell types, and developmental stages. Sets in which UNC5B exon 8 PSI values were not reported or quality scores do not meet the minimum threshold were excluded.

TSVdb web-tool (http://www.tsvdb.com)[112] was used to obtain UNC5B-Δ8 isoform-specific expression (not available with TCGA SpliceSeq) in colon adenocarcinomas of TCGA-COAD dataset (RSEM values, RNA-Seq by Expectation Maximization). Gene expression of NOVA2 and genes of the colon tumor-specific angiogenetic signature (CD59, COL4A1, HMGB1, COL4A1, HEYL, IGFBP7, PPAP2B, SPARC, ARPC2, LDHB, PHC3, POMP, VIM) were also retrieved with TSVdb. The tumor-specific angiogenetic signature was calculated as the average of normalized gene expression level (relative to the mean of expression in the TCGA-COAD dataset) of each gene of the signature.

**Survival analyses.** Overall survival curves of colon cancer patients were generated by analyzing clinical data (overall survival and survival status) in patients with colon adenocarcinoma of the TCGA-COAD dataset. Patients were stratified according to UNC5B-Δ8 and NOVA2 RSEM expression values in: High (>3rd quartile), Medium (between 3rd and 1st quartile), Low (<1st quartile). The log-rank Mantel-Cox test was employed to determine any statistical difference between the survival curves. Clinical data, UNC5B-Δ8 isoform-specific, and NOVA2 expression data were obtained from TSVdb (http://www.tsvdb.com). Metastasis-free and Relapse-free survival curves showed in Supplementary Fig. 14 were obtained by using the PROGgeneV2 tool (http://genomics.jefferson.edu/proggene)[113]. Patients were bifurcated based on the 75th percentile in high NOVA2 expression (>75th) or low NOVA2 expression (<75th).

**Identification of intrinsically disordered regions (IDR).** Intrinsically disordered regions in UNC5B protein sequence were retrieved by using the following bioinformatic tools: (i) GlobPlot prediction analysis (http://globplot.embl.de)[114] and (ii) PONDR prediction (http://www.pondr.com) using VLS2 algorithm optimized for long and short disordered regions[115]. Specifically, GlobPlot prediction analysis was performed using Russell/Linding propensity with the following parameters:

minimum peak width (5) and maximum join distance (4) in disorder prediction; minimum domain length (74) and maximum linker length (15) in globular domain hunting with SMART/Pfam domain prediction; smoothing settings were 10 for 1st and 2nd derivative Savitzky-Golay frame. Prediction of natural disordered regions (PONDR) was performed by using the VLS2 algorithm. PONDR VSL2 predictor is a combination of neural network predictors optimized for long (>30 residues) and short (≤30 residues) disordered regions.

**Statistics and reproducibility**. Paired or unpaired Student's two-tailed *t*-test and one-way ANOVA (corrected for multiple comparisons, Turkey test), were used to compare two or more groups, respectively, and to determine statistical significance (GraphPad Prism 6). When not differently indicated the mean value of each condition is shown. For survival analysis, the Log-rank test for trend was performed. Differences were considered significant at *P* value <0.05. When not differently indicated, three independent biological experiments were analysed (Fig.: 4b-c and 4e-f; Supplementary Figs.: 5b, 5c, 7c, 7f, 10b). Validation of BaseScope probe, TUNEL analysis, and Cleaved Caspase-3 IHC analysis was performed in parallel to the experimental conditions.

**Reporting summary**. Further information on research design is available in the Nature Research Reporting Summary linked to this article.

## Data availability

The previously published RNA-sequencing data of Nova2 knockdown and control ECs used in this study are available in the NCBI BioProject under accession code PRJNA293346 (ID: 293346)[27]. All gene and transcripts data, patient's survival, and clinical information that support the findings of this study are available in GDC Data Portal (https://portal.gdc.cancer.gov/projects/TCGA-COAD), ONCOMINE (htttp://www.oncomine.org), HCMDB databases (https://hcmdb.i-sanger.com), PROGgeneV2 tool (http://genomics.jefferson.edu/proggene), and TSVdb web-tool (http://www.tsvdb.com), VastDB (https://vastdb.crg.eu/wiki/Main_Page). Specific accession codes to original raw data used in this study include: GSE14095, GSE87211, GSE28722, GSE31595, and GSE28814. Source data are provided with this paper.

## Materials availability

Materials are available from the authors upon reasonable request.

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

## Acknowledgements

We are grateful to Dr. Chiara Mondello for providing mouse genomic DNA, Dr. Anna Garbelli for assistance with confocal microscope analyses, Dr. Jon D. Moulton for *i7_UNC5B* morpholino design, and Dr. Silvia Faravelli for support with HEK293-F cell cultures and transfections. We thank Dr. Davide Gabellini for the helpful discussion, and Dr. Giuseppe Biamonti, Dr. Alessandra Boletta, and Dr. Rosa Bernardi for the critical reading of the manuscript. We are also grateful to Prof. Elisabetta Dejana for help in performing experiments with ECs and critically reviewing the manuscript. We also thanks Dr. Giorgio Scita for pEN_Tmcs vector and Prof. Elena Seiradake for the pHLSec-UNC5B-FL vector. We acknowledge ARDIS Srl for provision of the Surface Plasmon Resonance instrumentation and the "Fondazione Adriano Buzzati-Traverso" for the support. Part of the results included in this work comes from data generated by the TCGA Research Network (https://www.cancer.gov/tc). This research was funded by Associazione Italiana per la Ricerca sul Cancro to C.G. (AIRC, projects IG-17395 and IG-21966). D.P. was supported by a AIRC fellowship for Italy. Research in F.F. lab was supported by the Armenise-Harvard Foundation (CDA 2013), by AIRC (MFAG 20075), and by the Italian Ministry of Education, University and Research (MIUR): Dipartimenti di Eccellenza Program (2018–2022, to the Dept. of Biology and Biotechnology "L. Spallanzani", University of Pavia). A.C. was supported by the European Union's Horizon 2020 research and innovation program under the Marie Curie grant agreement COTETHERS - n. 745934. This research was also funded by INCA (P.M.), Ligue contre le Cancer (P.M.), and ANR-(17-CONV-0002 and 10-LABX-0061) (P.M.).

## Author contributions

D.P. designed and performed most of the wet experiments, performed the bioinformatic analyses, and wrote the manuscript; G.D. and Alex Pezzotta[1] performed experiments in zebrafish; A.D.M., E.B., and D.C. contributed to ECs RNA collection, RBFOX2 knockdown, and generate expression vectors for GFP-tagged or HA-tagged UNC5B splicing isoforms; M. C., A.C., L.S., and F.F. purified recombinant UNC5B isoforms and designed SPR experiment; N.V., M.R., and S.Z. generated HUVECs knockdown for NOVA2, purified total RNA from cardiac ECs and critically reviewed the manuscript; Costanza Giampietro[2] generated ECs stable expressing UNC5B variants under the control of a tetracycline inducible promoter; M.P. P. performed CLIP experiments; Andrea Paradisi[2] and P.M. provided total RNA from tumors samples, complementary experience and reagents for analysis of the dependence receptor activity of UNC5B in ECs, and critically reviewed the manuscript; M.B. and W.V. performed all IHC and chromogenic BaseScope in situ hybridization analyses; Anna Pistocchi[3] provided help in performing zebrafish experiments and critically reviewed the manuscript; A.E. provided help in performing experiments with ECs and critically reviewed the manuscript; Claudia Ghigna conceived and designed the study and wrote the manuscript. All authors read and approved the manuscript.

## Competing interests

Claudia Ghigna is a consultant for Gene Tools. Funding bodies had no role in the design of the study and collection, analysis, and interpretation of data and in writing the manuscript. The remaining authors declare no competing interests.
