## [Peer Review File · Nature Communications]

REVIEWER COMMENTS

Reviewer #1 (Remarks to the Author):

The present study from Claudia Ghigna's lab investigated the biochemical feature of neurovascular guidance regulator, *Unc5b*, known to be highly context-dependent and functionally variable depending on the existence or absence of the ligand, *Netrin-1*. The authors identified a novel *Unc5b* splicing isoform generated through exon skipping by *NOVA2*. This isoform (*Unc5b-Δ8*), not regulated by *Netrin-1*, is constitutively pro-apoptotic and contributes to proper vascular development of fish embryos. Intriguingly, the expression of *Unc5b-Δ8* correlates with tumor angiogenesis and poor patient outcome in human colon cancers. Overall, the data promote our understanding of the mechanism of angiogenesis and the complicated action of *Unc5b*. This reviewer highly recommends this paper to be published in *Nature Communications* if the authors adequately address several points as listed below:

(Abstract: line 55)

"UNC5B-Δ8a" seems to be described as the name of proteins but not a short transcript. To avoid confusion, this should be corrected.

(Introduction: line 93)

The claim of Wilson et al. is that *Netrin1* functions as a pro-angiogenic factor rather than repulsive guidance cue for *Unc5b*. The description should be accurate at this point.

(Fig. 1c and Fig. S1b)

Why are the levels of *Unc5b-Δ8* in control cells remarkably different (32.1% vs. 0.1%) between these figures?

(Fig. 1d and Fig. S1c)

Are these the same experiments (si-Nova2 on luEC)? If not, please explain more.

(Fig.1g)

% values are incorrect.

(Fig. 2b)

The data indicates *Nova2* also binds the region B. Therefore the description "but not to control regions" in the line 244 should be modified. To conclude the direct binding of *nova2* onto intron 7, it will helpful to confirm that *nova2* does not bind mutated intron 7 YCAY motifs (used in Fig. 2d) by CLIP analysis.

(Fig.3c)

The difference is too small (97.6% vs 92.1%). The authors should show at least SEM or statistical evaluation.

(Fig. 3e-g)

It is confusing that *Unc5b-FL* could not rescue the defects in *Mo-unc5b* and *Nova2* mutants. If this is related to the dosage of injected RNA, authors should explain more or repeat the same experiments with higher dosage. Related to this, it is helpful to perform *UNC5B-d8*-specific knockdown by designing the antisense at junctions of exon 7 and 9.

(Fig. 4g)

Authors claim that pro-apoptotic activity of *UNC5B-d8* induces sprouting of PAV. Although results obtained by BAF inhibitor support this hypothesis, it's not clear how apoptosis supports EC sprouting. Authors should show clear images visualizing EC apoptosis or provide much clearer explanation.

Yoshiaki Kubota

Reviewer #2 (Remarks to the Author):

The authors describe the effects of a vascular specific isoform of *UNC5B* (*UNC5BΔ8*), which gets generated by

through Novo2 mediated alternative splicing. They claim that loss of exon 8 results in the inability to bind Netrin-1 and therefore renders the protein pro-apoptotic. The authors make their claims using a variety of models from tissue culture to in vivo.

A major issue with the paper is the meaning of bits of collected data. How does the endothelial specific pro-apoptotic signal drive angiogenesis or promote tumor metastasis and poor patient survival?

The authors claim that in the zebrafish model Unc5b deficiency leads to loss of parachordal formation by inhibition of apoptosis, which can be rescued by reintroducing Unc5b Δ 8, but not by full length Unc5b. However, there are a number of insufficiently analysed aspects regarding the zebrafish data: If Unc5b regulates parachordal formation by linking it so specific apoptosis of endothelial cells, the authors need to show when and where that apoptosis is occurring! just a general apoptosis inhibitor is not sufficient. Especially since specific apoptosis markers did not seem upregulated in nova2MO injected embryos (Giampietro et al. 2015; although this might depend on focussing on the right time point and region). It also would require a more refined model of how apoptosis contributes to angiogenesis of the PAV.

As there are no overview images of wt and mutant embryos, is it guaranteed that there are no global effects caused by global unc5b mutation? What happens to the thoracic duct at 5dpf? (or at 6dpf)?

Figure 3g show analysis of the parachordal at allegedly 6 dpf. Why did the authors choose such a later timepoint, especially as mRNA is only stable for less than 24 hours? Why is there no thoracic duct in the image?

In the image it seems, as if the DA is smaller or even collapsed in WT. This would be in agreement with the published nova2MO data. Or did the authors maybe not take the images at the exact same magnification (which also depends on the distance from objective and therefore on the mounting procedure)? In fig 3f wt and nova mutant do not seem to differ in their DAs. However, none of the described phenotypes are in agreement with the previously described effects of nova2 deletion. Please include experimental analysis of respective matching tissue.

In terms of function of Unc5b Δ 8:

How do the authors explain that also FL Unc5b can rescue parachordal formation, if provided in higher amounts (220pg)? (Wang et al 2009), especially since the authors do not use the published minimal dose for rescue (in that paper at 30pg), but also a high amount of 100pg?

Unfortunately the authors forgot to state the amount of Morpholino used:

As Unc5b acts as a dimer, does Unc5b Δ 8 act as a dominant negative? What is the effect of overexpression in Unc5b expressing endothelial cells or in vivo?

The combination with the tumor data seems arbitrary and not well explained

The authors show an upregulation of NOVA2 and UNC5B Δ 8 in colorectal tumor tissue. However, the interpretation or consequences seem unclear. In line 456-459 the authors state: "High NOVA2 expression is also associated with shorter metastasis and relapse-free survival (Supplementary Figure S7f-g). Finally, we found that high NOVA2 and UNC5B- Δ 8 expression levels correlated with shorter overall survival in colon cancer patients (Figure 5i-l)".

Why would the expression of a Pro-apoptotic signal in the tumor tissue negatively affect the patient??? Are other tumor types affected similarly?

How do the authors envision the use of NOVA2 as a prognostic marker, if one needs to isolate tumor vasculature to detect it? (at which state probably the tumor tissue will give more information about the prognosis...

please refrain from these generalized statements, which discredit exact science.

minor comments

Zebrafish strains need proper allele designations. Which alle are the authors referring to, when using the transgenic line Tg(Karl:GFP)? Please use the appropriate reference for the line! According to Zfin la116 refers to a kdrl:GFP, not kdr:GFP! As kdr is another Vegf receptor gene, it is important to be correct, these inaccuracies are very annoying and give the impression of sloppy science.

In general, please adhere to nomenclature guidelines (i.e. genes and promoters are small letters and italics, correct e.g. in lines 688, 695,720,799, 903.....; whereas proteins start with a capital letter, correct e.g. in line

295, 300, 309.....)

There are numerous spelling mistakes, including, but not limited to
line 411 Co-immunoprecipitatio misses an „n“
line 415 phosphorylation is misspelled

Reviewer #3 (Remarks to the Author):

In this study, Pradella et al. examined the involvement of a *Unc5b* isoform in regulating apoptosis and angiogenesis. The authors demonstrated that NOVA2 splicing factor promotes *Unc5bA8* isoform production in endothelial cells and in zebra fish embryos. The authors compared UNC5BFL and UNC5BA8 in their activity to induce cell death and to promote PVA vessel formation in zebra fish. Interestingly, the two isoforms displayed distinct effects in both biological processes. In addition, the authors found that NOVA2 and *Unc5bA8* upregulation correlated with more aggressive colon cancer progression.

The study is of relevance and interest to both basic developmental biology and to cancer research. However, there are some major issues that significantly weaken the validity of the authors' conclusion.

1. The authors demonstrated that UNC5BFL and UNC5BA8 had different abilities to induce cell death in response to Netrin and to rescue PAV formation reduced by *Unc5b* morpholino treatment. However, the link between the pro-apoptotic and pro-angiogenic activities of *Unc5b* is rather weak. The only in vivo evidence is the observation that BAF inhibitor, a pan-caspase inhibitor blocked the rescue by *Unc5bA8*.

A previous study by Wang et al., (Molecular Cell, 2009) shows that UNC5BAZU5 and UNC5BADD deletion mutations both increase apoptosis. However, these two mutants have distinct abilities to rescue PAV formation resulting from *Unc5b* knockdown. *Unc5bAZU5* injection is able to partially rescue, whereas *Unc5bADD* is unable to rescue. These results suggest that the pro-apoptotic activity of UNC5B does not equal its pro-angiogenic activity. Therefore, it is crucial to establish a direct and unambiguous link in *Unc5bA8*'s ability to induce cell death and to promote angiogenesis. An alternative model is that UNC5BA8 has a distinct guidance or branching activity than UNC5BFL.

Along the same line, DAPK (death associated protein kinase) has both caspase dependent and independent functions. The latter function involves regulation of autophagy. Thus, phosphorylation of and association with DAPK by UNC5B do not necessarily result in changes in apoptosis. Taken together, additional in vivo assays are needed to confirm that UNC5BA8 regulates angiogenesis through controlling apoptosis in vivo. For example, it would be informative to determine when and where apoptosis occurs in vivo during PAV formation and how distinct UNC5B isoforms affect these events.

2. Related to the first issue, the difference between the rescuing abilities between *Unc5bFL* and *Unc5bA8* could result from different expression levels of the protein products in vivo. Although the mRNAs were injected at the same amount (100 pg), they may not be expressed at comparable levels. This has been demonstrated previously in Wang et al.,

2009. Therefore, it is important to examine the rescuing effects of both isoforms at varying doses and compare the protein expression levels in injected animals. Furthermore, as UNC5BFL still had some pro-apoptotic activities in the presence of Netrin 1 (Figure 4d), why didn't it show any rescuing ability? As discussed above, this could suggest that the two activities are distinct from each other or this could result from different protein expression levels.

3. The conclusion of upregulation of NOVA2 protein and *Unc5b Δ 8* transcripts in primary tumors need to be further strengthened. First, in Figures 5 and S5, NOVA2 expression (in brown) seems be outside of CD31-positive (blue) EC as well, which is particularly evident in Cases #2 and #9. This seemed to be inconsistent with the notion that NOVA2 is restricted to EC. Second, the authors should specify all statistical analyses and p values in the figures or figure legends (e.g., Figures 5b,c,g,h and Figure S7 a,b,c,d). The reviewer found it almost impossible to evaluate the conclusions from Figures 5 and S7 without the original data sets.
4. In Figures 4 and S4, an intercellular GFP tag was used to examine surface/membrane localization of UNC5B isoforms. Although the signal was seen at cell periphery, this does not necessarily represent surface receptors that are capable of binding to Netrin. Instead, assays that specifically measure receptors with the extracellular domain displayed on the surface need to be performed, such as surface biotinylation assay or live antibody staining.

Minor issues:

1. The unique sequence in UNC5BFL is [356]NQRTLNDPKSHP[366] (12 a.a.), compared with [356]T (1 a.a.) in UNC5B Δ 8. The authors stated that UNC5B Δ 8 deletes residues Asn[356] to Thr[367], which include 12 a.a. This statement is inaccurate.
2. In Figure 5, instead of showing two examples of NOVA2 expression in tumor tissues, it would be more informative, in the reviewer's opinion, to include one tumor tissue and one non-tumor control (From Figure S5). And also include normal and tumor tissues for both *Unc5FL* and *Unc5b Δ 8* mRNAs in main Figure 5.
3. In Figures 5, S5, S6, it is confusing to use capitalized letter labels (e.g., A, B C) within panels that are labeled with small letters (e.g., a, b, c). Roman numerals may be better alternatives.

Point-by-point answers to the reviewer requests

Reviewer #1 (Remarks to the Author):

The present study from Claudia Ghigna's lab investigated the biochemical feature of neurovascular guidance regulator, *Unc5b*, known to be highly context-dependent and functionally variable depending on the existence or absence of the ligand, Netrin-1. The authors identified a novel *Unc5b* splicing isoform generated through exon skipping by NOVA2. This isoform (*Unc5b-Δ8*), not regulated by Netrin-1, is constitutively pro-apoptotic and contributes to proper vascular development of fish embryos. Intriguingly, the expression of *Unc5b-Δ8* correlates with tumor angiogenesis and poor patient outcome in human colon cancers. Overall, the data promote our understanding of the mechanism of angiogenesis and the complicated action of *Unc5b*. This reviewer highly recommends this paper to be published in Nature Communications if the authors adequately address several points as listed below:

(Abstract: line 55): "UNC5B-Δ8a" seems to be described as the name of proteins but not a short transcript. To avoid confusion, this should be corrected.

A1: We have clarified that *UNC5B-Δ8* is a constitutively pro-apoptotic isoform of the *UNC5B* gene and not a different protein by replacing "protein" with "splicing isoform" in line 57 (page 2) of the revised manuscript.

(Introduction: line 93): The claim of Wilson et al. is that Netrin1 functions as a pro-angiogenic factor rather than repulsive guidance cue for *Unc5b*. The description should be accurate at this point.

A2: We thank the Reviewer for his suggestion. We have removed Wilson et al. reference (page 3 line 104), which is mainly focused on the Netrin-1 proangiogenic role in the endothelium.

(Fig. 1c and Fig. S1b): Why are the levels of *Unc5b-Δ8* in control cells remarkably different (32.1% vs. 0.1%) between these figures?

A3: In Giampietro et al., 2015 (PMID: 26446569) we have demonstrated that NOVA2 expression levels in endothelial cells (ECs) are regulated by cell density. In particular, NOVA2 is significantly upregulated in confluent versus sparse ECs. We take advantage of this feature when we have to perform overexpression and knockdown studies. Hence, we usually perform NOVA2-overexpression in sparse ECs (in which control cells have relatively low NOVA2 levels). On the contrary, we usually perform NOVA2 knockdown in confluent ECs (in which NOVA2 control cells display relatively high NOVA2 levels). Since control cells in Fig. 1c and Fig. S1b have different NOVA2 expression levels, *UNC5B-Δ8* levels are also different. Actually, this is yet another evidence that NOVA2 and *UNC5B-Δ8* levels are tightly linked.

The different culturing conditions used to perform the experiments displayed in Fig. 1c and Fig. S1b are described in the methods section (page 16, lines 652-653). To better highlight the contribution of cell density and avoid confusion to the reader, in the revised manuscript, we have provided an additional supplementary figure (Supplementary Fig. 2a-c) showing NOVA2 expression levels and *UNC5B* exon 8 splicing analysis in moEC, lu2EC, and HUVEC/TERT2 grown as sparse or confluent. We have also included a short paragraph to address the effect of cell density on *NOVA2* and *UNC5B-Δ8* expression (page 5, lines 181-183).

(Fig. 1d and Fig. S1c): Are these the same experiments (si-Nova2 on luEC)? If not, please explain more.

A4: luEC (Fig. 1d) and lu2EC (Fig. S1c) are two different murine endothelial cell lines that were previously described in Magrini et al., 2014 (PMID: 25157817) and Bazzoni et al., 2005 (PMID: 15657074), respectively.

We are showing the effect of NOVA2 depletion on *Unc5b-Δ8* production in additional cell lines (in addition to moEC) to further support the NOVA2-dependent regulation of *Unc5b* splicing in ECs.

(Fig.1g): % values are incorrect.

A5: We apologize for the mistake. Correct % values have been provided in the revised manuscript.

(Fig. 2b): The data indicates Nova2 also binds the region B. Therefore the description “but not to control regions” in the line 244 should be modified. To conclude the direct binding of nova2 onto intron 7, it will be helpful to confirm that nova2 does not bind mutated intron 7 YCAY motifs (used in Fig. 2d) by CLIP analysis.

A6: We have revised the statement in line 244 as follows: " We found strong NOVA2 enrichment on the endogenous *Unc5b* pre-mRNA at the level of the YCAY cluster in intron 7 (Fig. 2b)." (page 6, lines 227-228).

To further demonstrate sequence specific association of NOVA2 to *Unc5b* intron 7 region – indicated by our bioinformatic predictions, CLIP on mouse ECs, and *in vitro* splicing assay –, we have performed RNA pull-down experiments by using riboprobes containing the YCAY cluster of *Unc5b* intron 7 region or the same region in which YCAY sites have been mutated to prevent NOVA2 binding.

As shown in Supplementary Fig. 4a and described in the main text (page 6, lines 240-244), NOVA2 is able to bind the RNA probe containing the identified YCAY cluster of *Unc5b* intron 7 while mutations of YCAY sites of the cluster drastically reduce NOVA2-binding.

Furthermore, in the revised manuscript we show that delivery of an antisense morpholino oligo annealing to the intronic YCAY cluster reduces *UNC5B-Δ8* production (Supplementary Fig. 10a-d) further supporting that NOVA2 controls *UNC5B-Δ8* production by binding to the YCAY cluster.

(Fig.3c): The difference is too small (97.6% vs 92.1%). The authors should show at least SEM or statistical evaluation.

A7: In the revised manuscript, we have provided the statistical evaluation of the partial rescue of *unc5b-Δ8* production by the co-injection of a morpholino-resistant *nova2* mRNA in *nova2*-morphants zebrafish embryos as supplementary figure (Supplementary Fig. 5a).

(Fig. 3e-g): **(I)** It is confusing that *Unc5b-FL* could not rescue the defects in *Mo-unc5b* and *Nova2* mutants. **(II)** If this is related to the dosage of injected RNA, authors should explain more or repeat the same experiments with higher dosage. **(III)** Related to this, it is helpful to perform *UNC5B-Δ8*-specific knockdown by designing the antisense at junctions of exon 7 and 9.

A8: In our opinion, *Mo-unc5b* and *nova2* mutants (*Tg(kdrl:GFP)la116;nova2io011*) zebrafish embryos provide two orthogonal models to investigate *unc5b-Δ8* biological activity *in vivo*. While *Mo-unc5b* allows comparing *Unc5b-FL* and *Unc5b-Δ8* activity in a *unc5b*-depleted background, in *nova2* mutants endogenous *Unc5b-FL* is present and only *Unc5b-Δ8* is depleted.

(I) Importantly, the ability of *Unc5b-Δ8* to restore PAV defects in *nova2* mutants (where endogenous *Unc5b-FL* is present) indicates that *unc5b-Δ8* and not *unc5b-FL* is responsible for the correct PAV formation. This result is further confirmed by the injection of *unc5b-FL* and *unc5b-Δ8* mRNAs in *unc5b*-depleted embryos, where *unc5b-FL* is not able to restore PAV. Collectively our results support the key role of *Unc5b-Δ8* in the formation of the

PAV during zebrafish development. We have better explained this concept in the revised manuscript (page 7 lines 295-319).

(II) The only reference in the literature of an attempt to restore PAV formation in *unc5b*-depleted embryos is the work performed by Wang and colleagues (Wang et al., 2009; PMID: 19328064) by using rat *Unc5b* constructs. Differently from Wang and colleagues, which injected different amounts of mRNA to compensate for the lower stability of mutated proteins, we use the same doses of *unc5b-FL* and *unc5b-Δ8* because we do not appreciate different protein expression levels between the two constructs. In the revised manuscript we have included a supplementary figure showing the same expression levels of zebrafish *unc5b-FL* and *unc5b-Δ8* constructs, as shown in Wang et al., 2009 for the different *unc5b* mutant constructs (Supplementary Fig. 5c).

To better evaluate the dose-dependence rescue of PAV defects, in the revised manuscript, we have injected increasing doses (100 pg; 150 pg; 200 pg) of *unc5b-FL* mRNA in *unc5b*-depleted zebrafish embryos. While for *unc5b-Δ8* we obtained a rescue of PAV formation already with low doses (50 pg), the highest dose of *unc5b-FL* was not able to full recover PAV formation (Supplementary Fig. 5d-e). Nevertheless, an increase in ECs PAV sprouting was detectable in embryos injected with the 200 pg dose of *unc5b-FL* mRNA (Supplementary Fig. 5f). The recovery of PAV upon forced expression of high doses of Unc5b-FL isoform observed by Wang and colleague and by us (ECs PAV sprouting only) suggests that increasing amounts of the Unc5b-FL isoform may mimic Unc5b-Δ8 activity. A possibility is that the Netrin-1 ligand, which is finely tuned during zebrafish development, could not completely prevent the pro-apoptotic signal of unbound Unc5b-FL receptors when they are overexpressed.

Notably, important differences with the work of Wang and colleagues are the use of zebrafish constructs instead of rat construct and the use of two different zebrafish strains (*Tg(kdrl:GFP)* and *Tg(fli1a:GFP)^{+/1}*), which may contribute to the lack of complete rescue of PAV formation by Unc5b-FL isoforms in *kdrl:GFP* zebrafish embryos.

(III) Finally, to better characterize UNC5B-Δ8 function, we have also performed *UNC5B-Δ8*-specific knockdown in human ECs (thus mimicking *nova2* mutants in which endogenous *unc5b-FL* is present and only *unc5b-Δ8* is depleted) by using two different approaches: *i*) short interfering RNA (siRNA); *ii*) morpholino antisense oligonucleotides.

By using a specific siRNA targeting human *UNC5B* exon 7-9 junction, we were able to deplete *UNC5B-Δ8* production in human ECs. Notably, *UNC5B-Δ8* depleted ECs showed a reduction in tube formation capabilities – measured as the number of segments and nodes formed – during *in vitro* angiogenesis, as evaluated by a tube formation assay on Matrigel (Fig. 4h).

Importantly, treatment of human ECs with a morpholino oligo blocking *UNC5B-Δ8* production was able to recapitulate tube formation defects observed upon *UNC5B-Δ8* siRNA-mediated depletion without affecting *UNC5B-FL* isoform and *UNC5B* total expression levels (Supplementary Fig. 10a-d).

Our novel results further support our conclusion that UNC5B-Δ8 has a key role in EC biology and during the angiogenic process.

(Fig. 4g): Authors claim that pro-apoptotic activity of UNC5B-d8 induces sprouting of PAV. Although results obtained by BAF inhibitor support this hypothesis, it's not clear how apoptosis supports EC sprouting. Authors should show clear images visualizing EC apoptosis or provide much clearer explanation.

A9: To support the role of apoptosis in the endothelium in physiological and pathological conditions, we have carried out a series of experiments in zebrafish embryos and tumor specimens.

In particular, we have investigated EC death during zebrafish development by TUNEL and Cleaved Caspase-3 immunostaining at different developmental stages (30 hpf, 33 hpf, 36 hpf, 48 hpf, 52 hpf). Despite detecting EC apoptosis is challenging for several reasons (i.e. apoptotic ECs are rapidly cleared by macrophages or moved at distant sites by blood flow), we were able to detect apoptotic ECs in the zebrafish trunk before PAV formation occurs. In particular, we noticed a small fraction of apoptotic ECs in the PCV region at 30 hpf-33 hpf, a developmental stage short before a second wave of sprouting emerges from the PCV to remodel ISVs and form the PAV that occurs at 34 hpf. Later, apoptotic ECs were sporadically detectable along the route of migrating ECs forming the PAV, including in ISV (Supplementary Fig. 9a-c). Thus, we were able to demonstrate that EC apoptosis occurs during zebrafish development and that apoptotic ECs are detectable before the beginning of the angiogenic wave that gives rise to a specific vascular bed (PAV).

We have also investigated the presence of apoptotic ECs in colon cancer tissues, where we were able to detect NOVA2 positive ECs (NOVA2+ and CD31+) immunolabeled with an anti-cleaved Caspase 3 antibody (Supplementary Fig. 15a-b). Notably, apoptotic ECs were extremely rare in the normal vasculature, suggesting that the presence of dying ECs may represent an important feature of the tumor vasculature.

In the revised manuscript, we have discussed the possible biological functions of EC death for tumor progression (page 15 lines 610-614).

Collectively, our novel analyses are in line with a relevant biological role of EC apoptosis during normal development and tumor progression. While, we recognize that further studies are required to better assess the contribution of EC apoptosis in specific biological processes (i.e. metastasis spread), we think that our work may stimulate novel investigations of apoptotic signaling pathways regulating EC biology.

Reviewer #2 (Remarks to the Author):

The authors describe the effects of a vascular specific isoform of UNC5B (UNC5BΔ8), which gets generated by through Novo2 mediated alternative splicing. They claim that loss of exon 8 results in the inability to bind Netrin-1 and therefore renders the protein pro-apoptotic. The authors make their claims using a variety of models from tissue culture to in vivo.

A major issue with the paper is the meaning of bits of collected data. How does the endothelial specific pro-apoptotic signal drive angiogenesis or promote tumor metastasis and poor patient survival?

A10: In the revised manuscript, we have better discussed the role of pro-apoptotic signal in driving angiogenesis and tumor metastasis (page 12 lines 496-502 and page 15 lines 608-614). Several works support the notion that EC apoptosis plays a fundamental role in vascular physiology and pathology. In addition to participating in blood vessel remodeling, EC apoptosis also contributes to the angiogenic process and vascular morphogenesis. Indeed, inhibition of EC apoptosis has been shown to impair vessel formation and lumen formation *in vitro* and *in vivo* (Segura et al., 2002, PMID: 12039865; Peters et al., 2002, PMID: 12204657; Tertemiz et al., 2005, PMID: 15564598). Different mechanisms have been proposed for apoptosis' role during the formation of novel blood vessels, including removal of wrongly placed ECs, promotion of angiogenic sprouting of adjacent vessels

through plasma membrane hyperpolarization, or through engulfment of apoptotic EC debris that in turn lead to activation of pro-angiogenic signals (Weihua et al., 2005, PMID: 16357162).

Recently, the importance of EC death has emerged as an important mechanism hijacked by cancer cells to disseminated at distant sites. Indeed, different studies have shown the ability of tumor cells to induce EC apoptosis to pass the EC layer of blood vessels (Heyder et al., 2002, PMID: 12384796; Haskó et al., 2019, PMID: 31426859).

In the revised manuscript, we have provided further evidence that EC apoptosis takes place during normal vessel development and during tumor angiogenesis.

All in all, our data point to a novel pathway regulating EC apoptosis that is relevant for physiological and pathological angiogenesis. However, in the manuscript we have stated the limitations of our study related to the precise molecular mechanism connecting NOVA2/UNC5B- Δ 8 and induction of EC apoptosis to tumor cell spread at distant sites associated with poor patient survival. Nevertheless, we believe that our data may promote further research on this exciting area of tumor biology.

The authors claim that in the zebrafish model *Unc5b* deficiency leads to loss of parachordal formation by inhibition of apoptosis, which can be rescued by reintroducing *Unc5b Δ 8*, but not by full length *Unc5b*. However, there are a number of insufficiently analysed aspects regarding the zebrafish data:

If *Unc5b* regulates parachordal formation by linking it so specific apoptosis of endothelial cells, the authors need to show when and where that apoptosis is occurring!

A11: We have provided a detailed analysis of apoptotic events occurring in ECs during zebrafish development (see our answer A9 to Reviewer #1).

just a general apoptosis inhibitor is not sufficient.

A12: We have confirmed that apoptosis inhibition prevents *unc5b- Δ 8* rescue of PAV formation by using two additional apoptosis-specific inhibitors (Q-VD-OPh and Z-VAD-FMK). These results are included in the revised manuscript as supplementary figure (Supplementary Fig. 8g).

Especially since specific apoptosis markers did not seem upregulated in *nova2*MO injected embryos (Giampietro et al. 2015; although this might depend on focussing on the right time point and region). It also would require a more refined model of how apoptosis contributes to angiogenesis of the PAV.

A13: We agree with the Reviewer comment that detecting apoptosis in zebrafish embryos depends on focusing on the right time point and region. Detecting EC apoptosis in zebrafish embryos is challenging since apoptotic ECs are rapidly cleared by macrophages or moved at distant sites by blood flow. In Giampietro et al. 2015, cell death was evaluated only at 48 hp and on transversal sections. In this manuscript, we have investigated EC death during zebrafish development by TUNEL and Cleaved Caspase-3 immunostaining at different developmental stages (30 hpf, 33, hpf, 36 hpf, 48 hpf, 52 hpf) on longitudinal sections. We were able to detect apoptotic ECs in the zebrafish trunk before PAV formation occurs. In particular, we noticed a small fraction of apoptotic ECs in the PCV region at 30 hpf -33 hpf, a developmental stage short before a second wave of sprouting emerges from the PCV to remodel ISVs and forming PAV at 34 hpf. At later time points, apoptotic ECs were detectable only sporadically (Supplementary Fig. 9a-c).

As there are no overview images of wt and mutant embryos, is it guaranteed that there are no global effects caused by global *unc5b* mutation?

A14: In the revised manuscript we have provided brightfield images of *unc5b*-depleted zebrafish embryos (Supplementary Fig. 5d-e). Importantly, different groups have reported slightly different phenotypes upon *unc5b* depletion in zebrafish embryos. Since PAV defects were consistently reported by multiple and independent groups (Lu et al., 2004, PMID: 15510105; Navankasattusas et al., 2008, PMID: 18223200; Wang et al., 2009, PMID: 19328064), we have decided to focus our analysis on this particular vascular bed and to use PAV formation as a read-out of Unc5b isoform activity *in vivo*.

What happens to the thoracic duct at 5dpf? (or at 6dpf)?

A15: To better visualize the thoracic duct and evaluate specific defects in *unc5b* morphants, we have performed a morpholino mediated-knockdown of *unc5b* in *Tg(fli1a:GFP)^{y1}* zebrafish embryos. No defects of the thoracic duct were observed at 5 dpf.

Normal Thoracic Duct formation in *unc5b* morphants. Lateral views (fluorescence) of 5 dpf *Tg(fli1a:GFP)^{y1}* zebrafish embryos. Arrows indicate the thoracic duct. Analyzed embryos are indicated.

Figure 3g show analysis of the parachordal at allegedly 6 dpf. Why did the authors choose such a later timepoint, especially as mRNA is only stable for less than 24 hours?

A16: Even if PAV defects are visible at 52 hpf, when using *nova2* mutants we have decided to evaluate PAV formation at a later time point to unambiguously recognize *nova2* homozygous mutants from heterozygous and wild-type siblings. This experimental design allowed us to better classify the genetic background of analyzed embryos, thus providing us a more reliable evaluation of PAV defects and *unc5b-Δ8* rescue in these mutants. Furthermore, our analysis at later stages allowed us to assess that in both *unc5b* morphants and *nova2* mutants PAV formation is not retarded but completely abrogated.

Why is there no thoracic duct in the image?

A17: *nova2* mutant zebrafish have been generated in a *kdrl:GFP* background (Giampietro et al., 2015). Unc5b-depletion was also performed in the *kdrl:GFP* zebrafish strain. As reported by others (Jung et al. 2017; PMID: 28506987), cells of the thoracic duct do not express the *kdrl:GFP* reporter.

In the image it seems, as if the DA is smaller or even collapsed in WT. This would be in agreement with the published nova2MO data. Or did the authors maybe not take the images at the exact same magnification (which also depends on the distance from objective and therefore on the mounting procedure)?

In fig 3f wt and nova mutant do not seem to differ in their DAs. However, none of the described phenotypes are in agreement with the previously described effects of nova2 deletion. Please include experimental analysis of respective matching tissue.

A18: In our novel manuscript we have described and focused our attention only in PAV defects of *nova2* mutants. This phenotype was not deeply investigated in Giampietro et al., 2015 due to the fact that the characterization of vascular defects of *nova2*-morphants and *nova2* mutants zebrafish embryos were analyzed at 48 hpf, whereas complete PAV formation occurs later during development.

All images were taken at the same magnification by adopting the same mounting procedure. We recognize that DA enlargement in *nova2* mutants is more evident in Fig. 3g in comparison to Fig. 3f, however we respectfully disagree with the Reviewer statement that in Fig. 3f wt and *nova2* mutant do not seem to differ in their DA. Digital quantification of DA in *sibling-like* wt and *nova2* mutants shown in Fig. 3f indicates an enlargement of approximately 12%. Notably, a similar enlargement was also observed in 48 hpf *nova2* mutants described in the Supplementary Figure 11E (similar magnification of Fig. 3f) of Giampietro et al., 2015.

To better evaluate DA enlargement, and PCV alterations, we included additional representative images of the trunk of *nova2* mutant zebrafish embryos injected with *unc5b-FL* and *unc5b-Δ8* mRNAs as supplementary figure (Supplementary Figure 6b). Notably, even if Unc5b-Δ8 is able to restore PAV formation in *nova2* mutants, DA enlargement is not prevented, further supporting the importance of Unc5b-Δ8 in specific vascular beds.

In terms of function of Unc5bΔ8: How do the authors explain that also FL Unc5b can rescue parachordal formation, if provided in higher amounts (220pg)? (Wang et al 2009), especially since the authors do not use the published minimal dose for rescue (in that paper at 30pg), but also a high amount of 100pg?

A19: The experiments by Wang et al 2009 are not entirely comparable to ours. Nevertheless, to address this point we have repeated the rescue experiments by titrating various *unc5b-FL* and *unc5b-Δ8* mRNA quantities. Please, see our answer A8 point to Reviewer #1 comment for a discussion of the results.

Unfortunately the authors forgot to state the amount of Morpholino used:

A20: The amount of morpholino used for Unc5b knockdown in zebrafish embryos have been specified in the methods section (page 25 line 956-957).

As Unc5b acts as a dimer, does Unc5bΔ8 act as a dominant negative? What is the effect of overexpression in Unc5b expressing endothelial cells or in vivo?

A21: The characterization of Unc5b-FL/Unc5b-Δ8 dimer ability to activate Netrin-1 pro-survival signal requires further structural investigation to assess how the absence of the IDR encoded by exon 8 impacts the intracellular conformation of the dimeric receptor. Additionally, Unc5b-Δ8 could compete for ligand availability, thus behaving as a dominant-negative receptor. However, our data indicate that the "possible" dominant-negative

activity of Unc5b- Δ 8 is not required for its function *in vivo*. Indeed, Unc5b- Δ 8 restores PAV formation in the absence of Unc5b-FL (in *unc5b* morphants) or when Unc5b-FL is present (in *nova2* mutants).

Moreover, to better investigate the biological activity of Unc5b- Δ 8 *in vivo*, we have injected wild-type zebrafish embryos expressing Unc5b with increased doses of *unc5b- Δ 8* mRNA (Supplementary Fig. 6c-f). Overexpression of Unc5b- Δ 8 in a wild-type background causes several vascular defects in a dose-dependent manner, including loss of arterial-ISV formation abnormalities and vessel defects in the plexus region (Supplementary Fig. 6c-f) (page 7, lines 289-292). Our results suggest that the pro-apoptotic activity of Unc5b- Δ 8 is independent of Unc5b-FL expression and the pro-survival signal activated by Unc5b-FL:Netrin-1 interaction is not able to compensate the activation of a cell death program *in vivo*.

The combination with the tumor data seems arbitrary and not well explained. The authors show an upregulation of NOVA2 and UNC5B Δ 8 in colorectal tumor tissue. However, the interpretation or consequences seem unclear. In line 456-459 the authors state: "High NOVA2 expression is also associated with shorter metastasis and relapse-free survival (Supplementary Figure S7f-g). Finally, we found that high NOVA2 and UNC5B- Δ 8 expression levels correlated with shorter overall survival in colon cancer patients (Figure 5i-l)". Why would the expression of a Pro-apoptotic signal in the tumor tissue negatively affect the patient???

A22: In the revised manuscript, we have stressed the fact that the UNC5B- Δ 8 pro-apoptotic signal is restricted to ECs of the tumor vasculature. Indeed, tumor cells do not express either NOVA2 and UNC5B- Δ 8 splicing isoform. We have also better discussed the importance of a tightly controlled pro-apoptotic signal in tumor ECs by providing several examples of the involvement of EC death in tumor spread at distant sites (page 15 lines 610-614). Moreover, EC apoptosis is fundamental during different phases of developmental angiogenesis (discussed on page 12 lines 496-502). Thus activation of EC apoptosis in tumor ECs may reflect the activation of a tumoral angiogenic program. In support of this notion, we were able to detect apoptotic ECs in the tumor vasculature, whereas EC apoptosis in the normal vasculature represent an extremely rare event (Supplementary Fig. 15a-b).

Are other tumor types affected similarly?

A23: As suggested by the Reviewer, in the revised manuscript we have included an additional analysis of two other tumor types (hepatocellular carcinoma and oral cavity carcinoma) (Supplementary Fig. 12a-d). In both these two cancer types, we were able to confirm NOVA2 up-regulation in tumor ECs compared to ECs of non-pathological adjacent tissues. Notably, analysis of TCGA datasets (TCGA-LIHC and TCGA-HNSC) showed *NOVA2* and *UNC5B- Δ 8* up-regulation at mRNA level in tumor specimens as demonstrated in colon cancer.

We have already demonstrated that EC-restricted NOVA2 expression is up-regulated in the vasculature of ovarian cancers and NOVA2 overexpression is associated with poor prognosis in ovarian cancer patients (Angiolini et al., 2019). Finally, EC-restricted *NOVA2* expression levels are up-regulated in gastric cancers. Notably, in stomach adenocarcinomas, high NOVA2 expression levels are associated with the presence of metastatic lymph nodes and poor overall survival (manuscript in preparation).

Collectively, our data strongly indicate that NOVA2 up-regulation and, consequently, the production of NOVA2-dependent splicing isoforms are a common feature of the tumor vasculature.

How do the authors envision the use of NOVA2 as a prognostic marker, if one needs to isolate tumor vasculature to detect it? (at which state probably the tumor tissue will give more information about the prognosis... please refrain from these generalized statements, which discredit exact science.

A24: We have recently reported the pro-angiogenic activity of a novel NOVA2-regulated splicing isoform of the L1CAM receptor (Angiolini et al., 2019). We have shown that in ECs of ovarian cancer vasculature NOVA2 promotes the skipping of the exon encoding for L1CAM transmembrane domain, thus generating a soluble isoform that is released in the extracellular environment. Notably, cleaved and soluble L1CAM ectodomains can be detected in the sera of different cancer patients, including ovarian, gastrointestinal, and breast cancer patients, thus representing a potential biomarker detectable in blood specimens (van der Maten et al., 2019; PMID: 31455004). Furthermore, different function-blocking antibodies targeting L1CAM have been successfully used in preclinical tumor models (reviewed in Angiolini & Cavallaro; 2017; PMID: 28134764), thus providing a strong rationale for moving in cancer patients. Hence, we believe that the characterization of NOVA2-dependent splicing in different tumor types could point to novel diagnostic, prognostic or therapeutic options.

We apologize to the Reviewer if we have generalized too much our vision in the last sentence of our manuscript. However, due to the limited space of the discussion, we have not amplified this concept in the revised manuscript.

minor comments

Zebrafish strains need proper allele designations. Which allele are the authors referring to, when using the transgenic line Tg(Karl:GFP)? Please use the appropriate reference for the line! According to ZFIN la116 refers to a *kdr1:GFP*, not *kdr:GFP*! As *kdr* is another Vegf receptor gene, it is important to be correct, these inaccuracies are very annoying and give the impression of sloppy science.

In general, please adhere to nomenclature guidelines (i.e. genes and promoters are small letters and italics, correct e.g. in lines 688, 695, 720, 799, 903.....; whereas proteins start with a capital letter, correct e.g. in line 295, 300, 309.....)

A25: We have carefully revised zebrafish strain nomenclature. *Tg(kdr1:GFP)la116;nova2io011* strain has been properly referred in the methods section.

For the nomenclature of genes and proteins, we used the following references: HUGO Gene Nomenclature Committee (HGNC); International Committee on Standardized Genetic Nomenclature for Mice; ZFIN Zebrafish Nomenclature Conventions.

There are numerous spelling mistakes, including, but not limited to

line 411 Co-immunoprecipitation misses an „n“

line 415 phosphorylation is misspelled

A26: We thank the reviewer. We have also carefully revised the entire manuscript to correct spelling mistakes.

Reviewer #3 (Remarks to the Author):

In this study, Pradella et al. examined the involvement of a *Unc5b* isoform in regulating apoptosis and angiogenesis. The authors demonstrated that NOVA2 splicing factor promotes *Unc5bΔ8* isoform production in endothelial cells and in zebra fish embryos. The authors compared *UNC5BFL* and *UNC5BΔ8* in their activity to

induce cell death and to promote PVA vessel formation in zebra fish. Interestingly, the two isoforms displayed distinct effects in both biological processes. In addition, the authors found that NOVA2 and Unc5b Δ 8 upregulation correlated with more aggressive colon cancer progression.

The study is of relevance and interest to both basic developmental biology and to cancer research. However, there are some major issues that significantly weaken the validity of the authors' conclusion.

1. The authors demonstrated that UNC5BFL and UNC5B Δ 8 had different abilities to induce cell death in response to Netrin and to rescue PAV formation reduced by Unc5b morpholino treatment. However, the link between the pro-apoptotic and proangiogenic activities of Unc5b is rather weak. The only *in vivo* evidence is the observation that BAF inhibitor, a pan-caspase inhibitor blocked the rescue by Unc5b Δ 8.

A27: We have better supported our finding that inhibition of apoptosis prevents Unc5b- Δ 8-dependent rescue of the PAV formation *in vivo* by using two additional apoptotic inhibitors (Q-VD-OPh and Z-VAD-FMK) (Supplementary Fig. 8g). Furthermore, to strengthen the link between the pro-apoptotic and the pro-angiogenic activities of UNC5B- Δ 8, we have performed an *in vitro* capillary-like tube formation angiogenesis assay (in which both pharmacological and genetic inhibition of apoptosis prevent the formation of a mature network, Segura et al., 2002 PMID: 12039865) in human ECs depleted of UNC5B- Δ 8 (Fig. 4h and Supplementary Fig. 10a-d).

A previous study by Wang et al., (Molecular Cell, 2009) shows that UNC5B Δ ZU5 and UNC5B Δ DD deletion mutations both increase apoptosis. However, these two mutants have distinct abilities to rescue PAV formation resulting from Unc5b knockdown. Unc5b Δ ZU5 injection is able to partially rescue, whereas Unc5b Δ DD is unable to rescue. These results suggest that the pro-apoptotic activity of UNC5B does not equal its pro-angiogenic activity. Therefore, it is crucial to establish a direct and unambiguous link in Unc5b Δ 8's ability to induce cell death and to promote angiogenesis. An alternative model is that UNC5B Δ 8 has a distinct guidance or branching activity than UNC5BFL. Along the same line, DAPK (death associated protein kinase) has both caspase dependent and independent functions. The latter function involves regulation of autophagy. Thus, phosphorylation of and association with DAPK by UNC5B do not necessarily result in changes in apoptosis. Taken together, additional *in vivo* assays are needed to confirm that UNC5B Δ 8 regulates angiogenesis through controlling apoptosis *in vivo*. For example, it would be informative to determine when and where apoptosis occurs *in vivo* during PAV formation and how distinct UNC5B isoforms affect these events.

A28: As illustrated by Wang et al., (Molecular Cell, 2009), while UNC5B Δ ZU5 and UNC5B Δ DD mutants both increase apoptosis, they do so to a very different extent: apoptosis is much stronger for UNC5B Δ ZU5 compared to UNC5B Δ DD. Accordingly, deletion of the DD domain nearly abolishes UNC5B ability to induce apoptosis when transfected in HEK-293T cells as shown in Llambi et al., 2001 (PMID: 11387206) and 2005 (PMID: 15729359). These results support the notion that EC apoptosis levels should be tightly controlled to regulate PAV formation *in vivo*.

While we cannot rule out the possibility that the absence of the IDR encoded by exon 8 affects UNC5B guidance or branching activity, the apoptotic phenotype was predominant in our analysis.

Furthermore, activation of autophagic markers (LC3B-I/LC3B-II conversion and ULK1 phosphorylation) is not differentially regulated in UNC5B-FL and UNC5B- Δ 8 overexpressing cells treated with Netrin-1, thus suggesting that DAPK phosphorylation and association with UNC5B receptor does not affect autophagy activation (included below).

Autophagic markers activation in UNC5B isoforms expressing cells. ULK phosphorylation (Ser757) and the autophagic marker Light Chain B (LC3B) conversion (LC3B-I/LC3B-II) in HeLa cells transiently transfected with UNC5B-FL-HA or UNC5B-Δ8 in 0,2% FBS and treated with 150 ng/ml of Netrin-1 (+) or BSA (-). α -TUBULIN as loading control. LC3B-I/LC3B-II ratio is also indicated.

To further prove *Unc5b-Δ8*'s regulation of apoptosis *in vivo*, we have included different experiments showing *Unc5b-Δ8* ability to induce EC death in wild-type and *unc5b*-morphants (see A9 and A21 answers). Notably, *unc5b* depletion reduces apoptotic ECs, which is restored by *unc5b-Δ8* mRNA injection together with PAV formation. As an additional control, we demonstrated that in *unc5b-Δ8-rescued* morphants treated with Caspase inhibitors the failure of PAV rescue is followed by a decrease of detectable EC death (Supplementary Fig. 8f). Altogether, our results establish a direct link between *Unc5bΔ8*'s ability to induce cell death and to promote angiogenesis.

Related to the first issue, the difference between the rescuing abilities between *Unc5bFL* and *Unc5bD8* could result from different expression levels of the protein products *in vivo*. Although the mRNAs were injected at the same amount (100 pg), they may not be expressed at comparable levels. This has been demonstrated previously in Wang et al., 2009. Therefore, it is important to examine the rescuing effects of both isoforms at varying doses and compare the protein expression levels in injected animals. Furthermore, as *UNC5BFL* still had some pro-apoptotic activities in the presence of Netrin 1 (Figure 4d), why didn't it show any rescuing ability?

A29: Wang and colleagues, by using rat *Unc5b* mutant constructs have shown that deletion of large intracellular domains (ZU5 or Death Domain) reduces protein stability thus lowering protein expression. In our case, zebrafish *Unc5b-Δ8* proteins show similar expression levels when the same amount of constructs is compared (Supplementary Fig. 5c).

Nevertheless, as requested by the Reviewer, we have repeated rescue experiments by titrating increasing amounts of *unc5b-FL* injected mRNA (see our answer A8 point I to Reviewer #1 comment).

The work of Wang et al., 2009 and our finding that at the highest *unc5b-FL* dose tested we found an increase in PAV sprouting further support the idea that induction of *Unc5b*-dependent apoptosis is required for the PAV formation. Alternative splicing, by generating an isoform insensitive to Netrin-1 pro-survival signal, provides a mechanism that does not require an increased *Unc5b* expression and could be controlled in a restricted cellular population (i.e. endothelial cells).

2. As discussed above, this could suggest that the two activities are distinct from each other or this could result from different protein expression levels.

A30: While we could not exclude the possibility that other *Unc5b*-downstream pathways may be affected by *Unc5b* exon 8 alternative as discussed in the revised manuscript (page 13 line 558 and page 14 lines 595-597), our data strongly support an intimate link between *Unc5b*- Δ 8 induction of apoptosis and PAV formation.

3. The conclusion of upregulation of NOVA2 protein and *Unc5bD8* transcripts in primary tumors need to be further strengthened. First, in Figures 5 and S5, NOVA2 expression (in brown) seems be outside of CD31-positive (blue) EC as well, which is particularly evident in Cases #2 and #9. This seemed to be inconsistent with the notion that NOVA2 is restricted to EC.

A31: To better prove that NOVA2 expression is restricted to ECs, in the revised manuscript we included an enlargement of CD31- and NOVA2- positive ECs (Supplementary Fig.11b) to clearly show the specific patten of NOVA2 expression. In particular, according to its role as splicing factor regulator, the NOVA2 signal is restricted to the nucleus. Differently, CD31 signal labels all the EC surface.

We respectfully disagree with Reviewer's interpretation of NOVA2 in Cases #2 and #9. NOVA2 true IHC signal is characterized by intense brown output restricted to the nucleus of the cell, whereas the blue signal of CD31 is diffused to the entire EC. The widespread low brownish signal in Cases #2 and #9 does not represent a true IHC signal, but instead is a background of the IHC reaction.

EC-restricted expression of NOVA2 in colon tissues has been previously demonstrated by using a different antibody by our group (Giampietro et al., 2015) and by another group (Gallo et al., 2018; PMID: 30275709).

Finally, we also provided a supplementary analysis of NOVA2 expression in additional tumor types, including hepatocellular carcinoma and oral cavity carcinoma, in which we showed that NOVA2 expression was restricted to ECs of tumor and non-pathological specimens (Supplementary Fig. 12a-d).

Second, the authors should specify all statistical analyses and p values in the figures or figure legends (e.g., Figures 5b,c,g,h and Figure S7 a,b,c,d). The reviewer found it almost impossible to evaluate the conclusions from Figures 5 and S7 without the original data sets.

A32: In the revised manuscript, according to *Nature Communications* editorial policy, we have included all exact P-values inside the figures and the description of the statistical analysis used in the figure legends.

Original data sets for transcript and gene expression levels used in Fig. 5 and 7 are freely accessible through the indicated web-resources:

- UALCAN web-tool in Fig. 5b, (<http://ualcan.path.uab.edu>);
- TCGASpliceSeq in Fig. 5c (<http://projects.insilico.us.com/TCGASpliceSeq>);
- TCGA-COAD dataset, including *NOVA2* expression levels and presence/absence of metastasis information, used in Fig. 5g was downloaded from cBioportal (<https://www.cbioportal.org>);
- Human Cancer Metastasis Database (HCMDDB) in Fig. 5h (<https://hcmdb.i-sanger.com>)
- Oncomine in Fig. S7 a,b (<http://oncomine.org/resource>);

To allow a better evaluation of our analysis, we have provided an additional supplementary table (Supplementary Table 3) including the following information related to Fig. 5, Supplementary Fig. 7 (Supplementary Fig. 14 in the revised manuscript), and Supplementary Fig. 12:

- Table summary showing NOVA2 low, q1, median, q3, and high expression levels in normal tissues and primary tumors analyzed by UALCAN (<http://ualcan.path.uab.edu>) related to Fig. 5b, Supplementary Fig. 12c, and Supplementary Fig. 12d.

- Raw data of PSI value in TCGA-COAD, TCGA-LIHC, and TCGA-HNSC datasets downloaded from TCGASpliceSeq and related to Fig. 5c, Supplementary Fig. 12c, and Supplementary Fig. 12d.
- Raw data of NOVA2 and UNC5B-Δ8 expression downloaded from TSVdb (<http://www.tsvdb.com>) related to Fig. 5d, 5e, and 5f.
- TCGA-COAD dataset showing NOVA2 expression levels and metastasis status related to Fig. 5g.
- NOVA2 and UNC5B-Δ8 expression levels quantified by RT-qPCR and RT-PCR, respectively, in colon cancer specimens related to Supplementary Fig. 14e.

Gene Expression Omnibus (GEO) Dataset IDs of datasets used in Fig. 5h and Fig. S7 a,b have been provided above each panel.

4. In Figures 4 and S4, an intercellular GFP tag was used to examine surface/membrane localization of UNC5B isoforms. Although the signal was seen at cell periphery, this does not necessarily represent surface receptors that are capable of binding to Netrin. Instead, assays that specifically measure receptors with the extracellular domain displayed on the surface need to be performed, such as surface biotinylation assay or live antibody staining.

A33: As requested by the Reviewer, we have performed a surface biotinylation assay. As shown in Fig. 4b and Supplementary Fig. 7a-d, both UNC5B-FL and UNC5B-Δ8 isoforms are able to reach the cell membrane and expose their N-terminal portion in the extracellular environment.

Minor issues:

1. The unique sequence in UNC5BFL is [356]NQRTLNDPKSHP[366] (12 a.a.), compared with [356]T (1 a.a.) in UNC5BΔ8. The authors stated that UNC5BΔ8 deletes residues Asn[356] to Thr[367], which include 12 a.a. This statement is inaccurate.

A34: We have revised the indicated statement to better clarify that skipping of *Unc5b* exon 8 lead to the production of an 11-aminoacid shorter protein isoform as follow: "... UNC5B-Δ8 (lacking amino acid sequence Asn^[356]-Pro^[367], which are substituted by a Thr^[367] residue), ..." (page 8 lines 348).

2. In Figure 5, instead of showing two examples of NOVA2 expression in tumor tissues, it would be more informative, in the reviewer's opinion, to include one tumor tissue and one non-tumor control (From Figure S5). And also include normal and tumor tissues for both *Unc5FL* and *Unc5bΔ8* mRNAs in main Figure 5.

A35: As requested by the Reviewer, we have included a tumor tissue and a paired normal-adjacent tissue for NOVA2 IHC analysis showed in the main Figure 5.

3. In Figures 5, S5, S6, it is confusing to use capitalized letter labels (e.g., A, B C) within panels that are labeled with small letters (e.g., a, b, c). Roman numerals may be better alternatives.

A36: We thank the Reviewer for her/his advice. To avoid confusion panels in Fig. 5 and Supplementary Fig. 11-12-13-15 have been labeled with roman numerals.

We thank all the Reviewers for their constructive comments and suggestions, which have allowed us to strongly improve our manuscript in different aspects.

The revised manuscript has been also revised to respond to editorial requests and formatting instructions.

In particular:

- Title has been shorted to 15 words.
- Abstract has been shorted to 150 words.
- Editorial policy checklist and reporting summary have been updated and provided as separated files.
- Bar graphs (when the sample size was < 10) have been replaced with plots that feature information about the distribution of the underlying data.
- Uncropped blots have been provided as supplementary figure (Supplementary Fig. 18).

REVIEWER COMMENTS

Reviewer #1 (Remarks to the Author):

In this revised paper, authors have adequately addressed my previous concerns and strengthened the data. Now the paper is acceptable.

Reviewer #2 (Remarks to the Author):

dear Authors

Thank you for your revised version. Most of my requests have been met. I do approve of the Apoptosis-assays and am well aware that it is extremely difficult to catch the apoptotic events affecting such a small cell population. I do not agree with your full interpretation of the results, however I am willing to accept different opinions on the contribution of apoptosis as a pro angiogenesis requirement. However I would like to caution you to phrase your comments a little less assertive given the weakness of the data.

Apart from that my only complain is that there is still no allele designation added to the Tg(kdrl:GFP).... (e.g. in Line 570, but also at all other places)

congratulations on your study

Reviewer #3 (Remarks to the Author):

The revised manuscript by Pradella et al. provided additional data to address issues such as in vivo apoptosis during angiogenesis and the dosage effect of Unc5b-FL in rescuing PAV vessel formation. However, the evidence provided together still does not fully support the key conclusions made by the authors, as outlined below.

1. Whether Unc5b- Δ 8 regulates angiogenesis through its pro-apoptotic activity is still unclear for the following reasons:

1a. By TUNEL and activated Caspase 3 staining, the authors demonstrated in vivo apoptosis in blood vessels (supplemental Figure 8,9). However, there appeared to be no difference in the apoptotic signals between treatment conditions at PAV (Supplementary Fig. 8f), where the effects of Unc5b isoforms were observed and quantified. Therefore, although apoptosis does occur at developing blood vessels, it does not explain the effects of Unc5b isoforms at PAV specifically.

1b. Although Unc5b- Δ 8 partially rescued PAV malformation caused by MO-Unc5b, Unc5b- Δ 8 overexpression blocked the formation of ISV (Supp Fig. 6). These opposite effects on blood vessel formation cannot be explained by the same pro-apoptotic activity of Unc5b- Δ 8.

1c. Although multiple apoptosis inhibitors were tested, the high concentrations used, 50 μ M for Q-VD-OPh hydrate and 300 μ M for Z-VAD-FMK, are likely to hit additional targets. The activities of these inhibitors on normal vessel formation on their own were not shown.

2. The conclusion that only Unc5b- Δ 8 has pro-angiogenic activity is not convincing for the following reasons:

2a. The authors still did not exclude the possibility that the different rescuing activities by Unc5b-FL and Unc5b- Δ 8 result from the two isoforms being expressed at different levels in vivo. Although the

authors showed comparable levels of the protein variants in HeLa cells (Supp Fig. 5), they did not examine it in embryos. It is possible that the half-life of Unc5b- Δ 8 is longer than that of FL in vivo and thus less Unc5b- Δ 8 mRNA is needed to maintain the same amount of protein expression during PAV development.

2b. In fact, the authors observed a dosage sensitive effect for Unc5b-FL in rescuing PAV formation (Supp Fig. 5d). Although the authors stated that the rescue was still not significantly different from MO-Unc5b at the highest dose tested, the controls used for Unc5b-FL (both MO-ctr and MO-Unc5b) were not the same as those for Unc5b- Δ 8 in Supp Fig. 5e. It is vitally important that the rescuing activities of the two isoforms be compared with the same set of positive and negative controls and in the same statistical analysis. In fact, the level of rescue by Unc5b-FL in Supp Fig. 5d was comparable to the rescue by Unc5b- Δ 8 in Supp Fig. 8g, with PAV segments across 10 ISV restoring to about 50% of the normal level under both conditions.

2c. Since Unc5b-FL has at least partial rescue activity, albeit requiring a higher dose, and Unc5b- Δ 8 also partially rescues PAV formation, it raised another possibility that a certain ratio between FL and Δ 8 is required to achieve the optimal pro-angiogenic activity. This ratio may be different for distinct vessels. This could potentially explain the authors' observation that overexpressing Δ 8 interfered with ISV formation.

3. Although there appears to be an upregulation of Nova2 and Unc5- Δ 8 in primary tumors, the existing evidence does not distinguish whether these molecules promote angiogenesis/tumorigenesis or vice versa. There is also significant activation of Caspase 3 outside blood vessels independent of Nova2 and Unc5- Δ 8 (Supp Fig. 15). Thus, poor survival of patients may be associated with additional causes.

Minor issues:

1. Is Unc5- Δ 8 exclusively expressed in ECs in normal tissues, such as the CNS?
2. Intron 7 sequences in Figure 2a were not displayed normally in the PDF file.

Point-by-point answers to the reviewer requests

Reviewer #1 (Remarks to the Author):

In this revised paper, authors have adequately addressed my previous concerns and strengthened the data. Now the paper is acceptable.

A: We thank the reviewer again for all his comments/suggestions that have allowed us to improve our manuscript drastically.

Reviewer #2 (Remarks to the Author):

dear Authors

Thank you for your revised version. Most of my requests have been met. I do approve of the Apoptosis-assays and am well aware that it is extremely difficult to catch the apoptotic events affecting such a small cell population. I do not agree with your full interpretation of the results, however I am willing to accept different opinions on the contribution of apoptosis as a pro angiogenesis requirement. However I would like to caution you to phrase your comments a little less assertive given the weakness of the data.

A: We thank the reviewer for her/his comments/suggestions that have encouraged us to better investigate endothelial cell apoptosis *in vivo*. In this version, we have carefully revised the manuscript to avoid any statement that is not fully supported by our data (see lines 388-390, 394-395, and 535-538). We also better specify that our observation of Unc5b- $\Delta 8$'s roles in the development of the vascular system could be restricted to particular vascular beds.

We hope that our interpretation of the results may stimulate the field to investigate further the contribution of endothelial cell apoptosis in the context of physiological and tumor angiogenesis.

Apart from that my only complain is that there is still no allele designation added to the Tg(kdrl:GFP).... (e.g. in Line 570, but also at all other places)

A: Allele designation has been added to the revised version of the manuscript.

congratulations on your study

Reviewer #3 (Remarks to the Author):

The revised manuscript by Pradella et al. provided additional data to address issues such as *in vivo* apoptosis during angiogenesis and the dosage effect of Unc5b-FL in rescuing PAV vessel formation. However, the evidence provided together still does not fully support the key conclusions made by the authors, as outlined below.

1. Whether Unc5b- $\Delta 8$ regulates angiogenesis through its pro-apoptotic activity is still unclear for the following reasons:

1a. By TUNEL and activated Caspase 3 staining, the authors demonstrated *in vivo*

apoptosis in blood vessels (supplemental Figure 8,9). However, there appeared to be no difference in the apoptotic signals between treatment conditions at PAV (Supplementary Fig. 8f), where the effects of Unc5b isoforms were observed and quantified. Therefore, although apoptosis does occur at developing blood vessels, it does not explain the effects of Unc5b isoforms at PAV specifically.

A: As properly stated by the Reviewer, by TUNEL and activated Caspase 3 immunostaining, we demonstrated that apoptosis normally occurs in blood vessels, as reported also by other authors in zebrafish embryos (Espín R. et al., 2013; PMID: 22956347). However, as reported by different studies, the PAV structure is fully detectable only from 48-52 hpf (Koltowska et al., 2015; PMID: 26655899).

Since apoptosis in endothelial cells temporarily occurs mainly before PAV formation, it is extremely difficult to simultaneously detect apoptosis in this structure. We have to point out that in Supplementary Fig. 8 and 9, we detect apoptotic signals in the region where the PAV would arise. Moreover, apoptosis was observed in the posterior cardinal vein (PCV) from which PAV originates. To clarify this point, we have modified the text as follow: "Notably, apoptotic cells were observed in the region of the PCV from which the PAV originates – at 30 hpf – or where the PAV would arise – at 33 hpf –." (see lines 323-324).

Furthermore, in the new version of the manuscript, we have included the number of embryos in which we have observed TUNEL/GFP or Cleaved-Caspase3/GFP co-localization signals for each condition. This information has been included in each representative image of Supplementary Fig. 8f, 9b, and 9c.

Please note that to precisely evaluate and quantify PAV formation in the different conditions analyzed, representative images of Supplementary Fig. 8f show zebrafish embryos at 52 hpf, a time point in which the transient apoptotic wave observed in the PCV region has already been passed. Nevertheless, we were able to demonstrate the ability of Unc5b- $\Delta 8$ to restore specific ECs apoptosis in the zebrafish trunk upon *Unc5b* depletion by comparing the TUNEL/GFP co-localization signal among the different experimental conditions.

1b. Although Unc5b- $\Delta 8$ partially rescued PAV malformation caused by MO-Unc5b, Unc5b- $\Delta 8$ overexpression blocked the formation of ISV (Supp Fig. 6). These opposite effects on blood vessel formation cannot be explained by the same pro-apoptotic activity of Unc5b- $\Delta 8$.

A: We respectfully disagree with the Review interpretation that Unc5b- $\Delta 8$ pro-apoptotic activity could not explain a dichotomy effect on blood vessels. A space- and time-controlled balance between pro-apoptotic and anti-apoptotic stimuli is required in different biological processes, including cardiovascular system development. In the revised version of this manuscript, we have better discussed our interpretation of these results. In particular, we have suggested that the tight control of Unc5b- $\Delta 8$ in the endothelium is required for specific vessel formation (PAV), whereas forced expression may be detrimental for other vascular structures, including the ISVs and vessels in the plexus region (see lines 268-270 and 484-487).

However, we have to underline that for the novel rescue experiments, we injected 50 pg of Unc5b- Δ 8 in a condition of depletion of Unc5b (with the injection of the *unc5b*-MO). This dose, in a wild-type background, did not induce severe defects in ISVs formation. Indeed, we detected blocked ISVs formation only when we overexpressed Unc5b- Δ 8 by injecting its mRNA at higher concentrations (from 75 to 100 pg) in a condition in which endogenous Unc5b is present. In our opinion, the two situations are different and cannot be compared in terms of Unc5b- Δ 8 effects on blood vessel formation. Moreover, we also discussed the possibility that different Unc5b- Δ 8/Unc5b-FL ratios among diverse vascular beds could be involved to achieve correct balance between pro-angiogenic and pro-apoptotic activities of the Unc5b pathway (see lines 498-506).

1c. Although multiple apoptosis inhibitors were tested, the high concentrations used, 50 μ M for Q-VD-OPh hydrate and 300 μ M for Z-VAD-FMK, are likely to hit additional targets. The activities of these inhibitors on normal vessel formation on their own were not shown.

A: The rationale for using the indicated concentrations of apoptosis inhibitors is based on previous works in zebrafish embryos (Castets et al., 2009 PMID: 18223200; Walters et al., 2008 PMID: 19097072; Goldsmith et al., 2013 PMID: 22917923; Gregory-Evans et al., 2011 PMID: 21677791; Williams et al., 2000 PMID: 11023682), where no obvious toxic side effect has been reported.

Furthermore, before performing the analyses with apoptotic inhibitors reported in the manuscript, we previously tested them in a dose-response assay in control embryos to exclude defects in blood vessel formation. These controls were not added to the revised manuscript in order not to make the supplementary information lengthy. The results are shown below in this point-by-point response. In addition, a comment on this has been added to the text: "Notably, inhibition of apoptosis prevented the rescue of *unc5b* morphants PAV formation defects by *unc5b*- Δ 8 mRNA without impairing other blood vessels (Fig. 4g, Supplementary Fig. 8g, and not shown)." (see lines 336-338).

Accordingly, in embryos treated with Q-VD-Oph and Z-VAD-FMK, we did not find any evident morphological defect in ISVs, DA, and PCV, as shown in Supplementary Fig. 8g and 8f.

Additional Figure. Q-VD-Oph hydrate and Z-VAD-FMK activities on normal vessel formation. *Tg(fli1a:GFP)^{yl}* zebrafish embryos were treated with different concentrations of Q-VD-Oph hydrate and Z-VAD-FMK apoptosis inhibitors, and ISV formation was evaluated. The percentage (%) of embryos with normal ISV is indicated. *Tg(fli1a:GFP)^{yl}* embryos treated with DMSO were used as control. The number of analyzed embryos is reported.

2. The conclusion that only Unc5b- Δ 8 has pro-angiogenic activity is not convincing for the following reasons:

2a. The authors still did not exclude the possibility that the different rescuing activities by Unc5b-FL and Unc5b- Δ 8 result from the two isoforms being expressed at different levels *in vivo*. Although the authors showed comparable levels of the protein variants in HeLa cells (Supp Fig. 5), they did not examine it in embryos. It is possible that the half-life of Unc5b- Δ 8 is longer than that of FL *in vivo* and thus less Unc5b- Δ 8 mRNA is needed to maintain the same amount of protein expression during PAV development.

A: We agree with the Reviewer and we thank her/him for the suggestion that Unc5b- Δ 8 and Unc5b-FL protein half-life could be different in distinct vascular contexts *in vivo*. We have included this possibility in the discussion section (see lines 498-506).

2b. In fact, the authors observed a dosage sensitive effect for Unc5b-FL in rescuing PAV formation (Supp Fig. 5d). Although the authors stated that the rescue was still not significantly different from MO-Unc5b at the highest dose tested, the controls used for Unc5b-FL (both MO-ctr and MO-Unc5b) were not the same as those for Unc5b- Δ 8 in Supp Fig. 5e. It is vitally important that the rescuing activities of the two isoforms be compared with the same set of positive and negative controls and in the same statistical analysis. In fact, the level of rescue by Unc5b-FL in Supp Fig. 5d was comparable to the rescue by Unc5b- Δ 8 in Supp Fig. 8g, with PAV segments across 10 ISV restoring to about 50% of the normal level under both conditions.

A: We understand the concern raised by the Reviewer. However, in all experiments in which we evaluated the ability of *unc5b* variants to restore PAV formation, MO-ctr and MO-*unc5b* embryos were always used as positive and negative controls, respectively. These controls are essential because the first one provides us information on the background/health of injected embryos, whereas the latter provides us a phenotypic readout of *unc5b* silencing. The experiments included in Supplementary Fig. 5d and Supplementary Fig. 5e were designed to answer two different biological questions: i) do higher doses of Unc5b-FL restore PAV formation of *unc5b* morphant zebrafish embryos? ii) does Unc5b- Δ 8 restore PAV formation also at a lower dose?

Please also note that in the specific experiments cited by the Reviewer, it is not entirely fair to only compare the restoring efficiency of each variant without considering the inhibition of PAV formation by Unc5b silencing, which was more pronounced in Supplementary Fig. 8g, therefore leading to a huge rescue.

Importantly, the side-by-side comparison of Unc5b-FL and Unc5b- Δ 8 was provided in Fig. 3e and Fig. 3g, where for each experiment, three independent biological replicates (embryos - at least 20 per condition - from different crosses and different batch of *unc5b* morpholino and *unc5b* mRNAs) were used. In the revised manuscript, as an additional control, we have also included the effect of Unc5b- Δ 8 in restoring the

PAV of MO-*unc5b* injected with the highest doses of *Unc5b-FL* mRNA (200 pg) (Supplementary Fig. 5f).

2c. Since *Unc5b-FL* has at least partial rescue activity, albeit requiring a higher dose, and *Unc5b-Δ8* also partially rescues PAV formation, it raised another possibility that a certain ratio between FL and Δ8 is required to achieve the optimal pro-angiogenic activity. This ratio may be different for distinct vessels. This could potentially explain the authors' observation that overexpressing Δ8 interfered with ISV formation.

A: This is a very intriguing possibility, which makes a lot of (biological) sense. We agree with the Reviewer's interpretation of our results. We have discussed this issue in the revised version of the manuscript (see lines 498-506). We thank the Reviewer for this important contribution.

3. Although there appears to be an upregulation of *Nova2* and *Unc5-Δ8* in primary tumors, the existing evidence does not distinguish whether these molecules promote angiogenesis/tumorigenesis or vice versa. There is also significant activation of Caspase 3 outside blood vessels independent of *Nova2* and *Unc5-Δ8* (Supp Fig. 15). Thus, poor survival of patients may be associated with additional causes.

A: We are aware that our data, despite providing a strong positive correlation between *NOVA2* and *Unc5-Δ8* expression with a pro-angiogenic cancer signature, do not directly demonstrate that *NOVA2* and *UNC5-Δ8* promote tumor angiogenesis *in vivo*. To directly test this, we generated conditional KO mice lacking *NOVA2* selectively in the endothelium. While this project just started and it will take one year to build solid data, the preliminary findings are in line with a positive role for *NOVA2* in tumor angiogenesis.

It is not unexpected to find apoptosis in tumor patient sections outside of blood vessels, as growing tumors display hypoxic/low nutrients areas that might lead to cancer cell apoptosis. It is also not surprising to find Caspase 3 positive cells independent of *NOVA2* and *UNC5-Δ8* since multiple pathways regulate apoptosis in tumors.

Hence, we have carefully revised the final manuscript to avoid any possible claims not fully supported by the included findings (see lines 388-390 and 534-538).

Minor issues:

1. Is *Unc5-Δ8* exclusively expressed in ECs in normal tissues, such as the CNS?

A: Differently from *Neogenin* and *DCC* netrin receptors, which were previously identified as *NOVA2* targets in the CNS (Saito et al., 2016; PMID: 27223325), to our knowledge *Unc5b* exon 8 splicing was not reported. Different possibilities may explain this observation, including different spatial- and time-restricted expression patterns of *NOVA2* and *Unc5b* splicing within the CNS.

2. Intron 7 sequences in Figure 2a were not displayed normally in the PDF file.

A: Figure 2a incorrect visualization has been fixed in the resubmitted version of this manuscript.